evolution/health and disease and epidemiology/theoretical biology

COVID-19, virulence evolution, vaccine

**Author for correspondence:**
Ian F. Miller
e-mail: ifmiller@princeton.edu

# Assessing the risk of vaccine-driven virulence evolution in SARS-CoV-2

Ian F. Miller[1] and C. Jessica E. Metcalf[1,2]

[1]Department of Ecology and Evolutionary Biology, and [2]Princeton School of Public and International Affairs, Princeton University, Princeton, NJ, USA

IFM, 0000-0002-2673-9618; CJEM, 0000-0003-3166-7521

The evolution of SARS-CoV-2 virulence, or lethality, threatens to exacerbate the burden of COVID-19 on society. How might COVID-19 vaccines alter selection for increased SARS-CoV-2 virulence? Framing current evidence surrounding SARS-CoV-2 biology and COVID-19 vaccines in the context of evolutionary theory indicates that prospects for virulence evolution remain uncertain. However, differential effects of vaccinal immunity on transmission and disease severity between respiratory compartments could select for increased virulence. To bound expectations for this outcome, we analyse an evo-epidemiological model. Synthesizing model predictions with vaccine efficacy data, we conclude that while vaccine-driven virulence remains a theoretical possibility, the risk is low if vaccines provide sustained robust protection against infection. Furthermore, we found that any increases in transmission concomitant with increases in virulence would be unlikely to threaten prospects for herd immunity in a highly immunized population. Given that virulence evolution would nevertheless impact unvaccinated individuals and populations with low vaccination rates, it is important to achieve high vaccination rates worldwide and ensure that vaccinal immunity provides robust protection against both infection and disease, potentially through the use of booster doses.

## 1. Introduction

Since its emergence in late 2019, the SARS-CoV-2 virus has spread globally. As of October 2021, it has resulted in over 243 million cases of its associated disease, COVID-19, and over 4.9 million deaths [1]. Beyond the morbidity and mortality associated with the pandemic, nations and sub-national states heavily affected by COVID-19 have experienced catastrophic economic collapse [2], critical pauses in educational services and pervasive psychological damage. Generating herd immunity through vaccination has been consistently identified as the only acceptable course of action for mitigating the pandemic and allowing society to begin the

recovery process. The race for SARS-CoV-2 vaccines has proceeded at an unprecedented pace. Following promising results from phase 3 trials [3–6] several vaccines have been authorized for use. While clinical trials provide the evidence necessary to confirm the ability of vaccines to safely prevent disease in individuals, potential population-level and longer-term consequences of vaccine introduction, particularly viral evolution, should also be considered. Potential negative outcomes include vaccine escape via antigenic evolution, which would result in a decrease or loss of vaccine efficacy [7], and the evolution of increased virulence, which could result in more severe health outcomes and a higher infection fatality rate in unvaccinated individuals. The recent emergence of the *alpha* and *delta* variants has drawn attention to the threat posed by SARS-CoV-2 virulence evolution. Some evidence indicates that the *delta* variant may be linked to increased infection fatality rates [8], although there is still considerable uncertainty [9,10].

In this paper, we assess the potential for vaccines to drive the evolution of SARS-CoV-2 virulence. First, we review the theory surrounding the evolution of virulence evolution as it relates to SARS-CoV-2. Next, we consider the effects of COVID-19 vaccines in this theoretical framework, evaluating current evidence and assessing the potential for selection for increased virulence using an evo-epidemiological model. Finally, we synthesize our findings and generate recommendations for minimizing this risk, a question of growing importance in the happy circumstance of ever greater deployment of vaccines globally.

## 2. The theory of virulence evolution

The evolution of pathogen virulence, which for the purposes of this paper we define as the rate of disease-associated mortality, has intrigued scientists for decades. In the absence of any associated costs, pathogens are expected to evolutionarily maximize both the rate at which they transmit from infected hosts, and the duration of time during which a host is infectious. Because mortality (and in some cases severe symptoms) curtails opportunities for transmission, virulence alone is never adaptive for pathogens, and is at best selectively neutral (e.g. when host mortality occurs after the infectious period). However, for many pathogens, host damage either can enhance transmission (e.g. by inducing coughing) or is an unavoidable consequence of or requirement for transmission (e.g. cellular damage resulting from viral replication). Thus, selection for increased virulence can occur as a result of selection for increased transmission [11–13], and indeed, only as a result of selection for increased transmission.

From this evolutionary theory, it is evident that two conditions must be met for virulence evolution to occur. First, increased transmission must be biologically feasible, and must be determined by the genetics of the pathogen (e.g. it must be heritable). Second, evolutionary increases in transmission must lead to increases in virulence (e.g. virulence must be an unavoidable consequence of transmission). These two conditions determine whether virulence evolution is possible, but they are not informative about the limits to virulence evolution, i.e. how destructive the pathogen has the potential to become, in terms of host mortality.

There are two major sets of factors that limit virulence evolution. The first involves limits to transmission evolution. Because virulence can only be selected for in association with transmission, any biologic, metabolic or other limits to transmission evolution (such as limits to the rate of protein synthesis or viral assembly) necessarily constrain virulence evolution [14,15]. The second set of factors limiting virulence evolution encompasses the effects of host mortality on transmission rate. If neither symptoms nor disease-associated death shorten the duration of transmission, then virulence does not reduce pathogen fitness and its evolution is not constrained [16] (figure 1, yellow lines). Conversely, if symptoms or disease can truncate the infectious period, then the potential for a trade-off between virulence and transmission that constrains the evolution of these two traits emerges.

To put this formally, for many pathogens, incremental increases in transmission are hypothesized to bear increasing costs of damage, leading to a positive, saturating relationship between mortality (or severity) rate and the maximum attainable transmission rate. This relationship—the canonical 'virulence transmission trade-off'—has the effect of limiting the evolution of virulence (figure 1, purple lines). The saturating nature of this trade-off curve makes an intermediate degree of virulence evolutionarily optimal, as the benefits of increased transmission are balanced against the costs of host death (or other severe outcomes truncating transmission). If the trade-off curve does not saturate (figure 1, teal lines), then any decreases in transmission time due to increased virulence are more than compensated for by increases in transmission rate, and the evolution of virulence is not constrained. It is important to note that the trade-off curve bounds the set of possible virulence and transmission

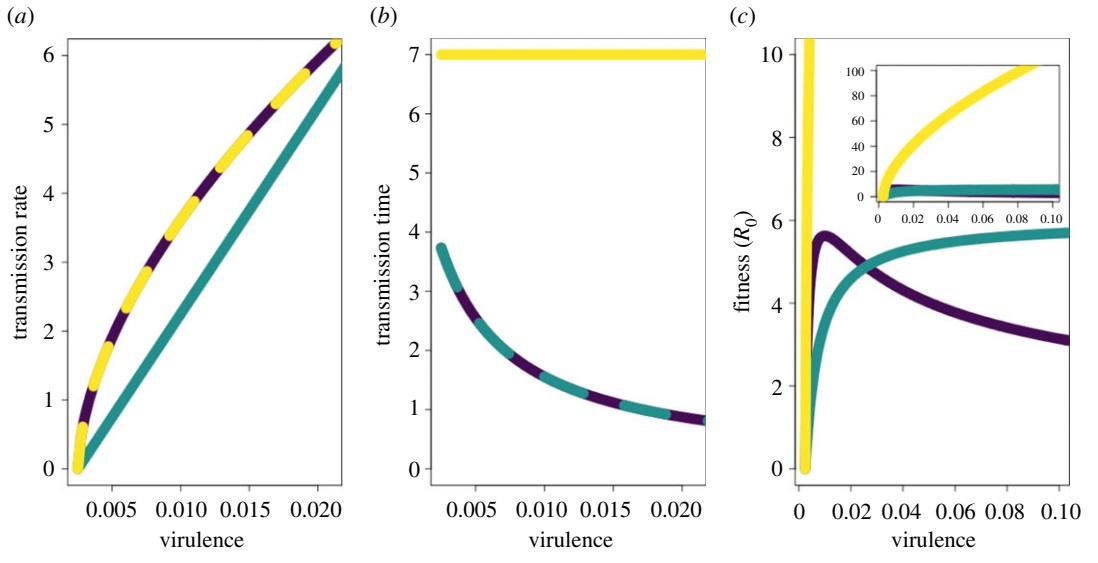

**Figure 1.** Virulence–transmission trade-offs. (*a*) Different possible shapes for the relationship between transmission and virulence. The curves define the maximum transmission rate attainable for a given virulence strategy (interpretable as, for example, the infection fatality ratio). The region under each curve represents the set of all possible combinations of transmission rate and virulence for that trade-off shape. (*b*) The relationship between virulence and transmission time. (*c*) The relationship between virulence and fitness (in a completely susceptible population, equivalent to $R_0$). Purple lines show relationships consistent with a virulence–transmission trade-off. Teal lines illustrate a non-saturating relationship between virulence and transmission. Yellow lines correspond to the scenario in which virulence does not decrease transmission time (for example, if mortality occurs after the infectious period has ended).

strategies. Evolutionary changes can occur on the interior of this set with unconstrained increases and/or decreases in both virulence and transmission, and as such the presence or shape of a trade-off may not be apparent until transmission is maximized for a given virulence strategy and evolution begins to 'trace the curve'. Direct evidence for the existence of virulence–transmission trade-offs is limited, but suggestive evidence has been found in many systems [13].

These theoretical frameworks can be used to assess whether the evolution of increased SARS-CoV-2 virulence is possible and to identify factors limiting the extent to which this could occur.

## 3. Evaluating potential for virulence evolution in SARS-CoV-2

The evidence surrounding differences in transmission and virulence between the ancestral SARS-CoV-2 strain and the emergent *alpha* (i.e. *B.1.1.7*) and *delta* (*B.1.617.2*) variants sheds light on whether the two conditions necessary for SARS-CoV-2 virulence evolution are met. These variants represent separate branches of viral evolution stemming from the ancestral strain [17,18]. The *alpha* variant was designated as a variant of concern by the World Health Organization in December 2020, and the *delta* variant received the same designation in May 2021 [19].

Current evidence surrounding these two variants gives a clear indication that the first condition (the feasibility and heritability of increased transmission rate) is satisfied. The *alpha* variant has been identified as being 43–82% more transmissible than ancestral genotypes [20]. An even more dramatic increase in transmission has been observed in *delta*, which is estimated to be at least twice as transmissible as *alpha* [21,22], and has become the predominant variant worldwide [1]. Other variants and mutations have also been linked to increased transmission. For example, the *D614G* mutation is hypothesized to be associated with increased transmission due to the rapid increase in its frequency [23,24], enhanced replication *in vitro* [24,25] and increased viral titers *in vivo* [26].

The second condition (virulence being a consequence of increased transmission) necessary for virulence evolution is similarly supported by patterns of increased mortality associated with the *alpha* and *delta* variants. The *alpha* variant is associated with an approximately 55% increase in infection fatality rate over the ancestral strain [27], and the *delta* variant has been linked to even greater increases in virulence over *alpha*. A retrospective cohort study examining records from Ontario,

Canada, found that infection with *delta* strain was associated with a 133% increase in the risk of death compared to infection with the ancestral strain [8]. Likewise, an observed increase in the risk of hospitalization in patients infected with the *delta* versus *alpha* strain [28] further points to an incremental increase in virulence in *delta*. However, several studies cast doubt on claims that *delta* is more virulent than *alpha*. In the United States, the proportion of children hospitalized with COVID-19 that went on to experience severe outcomes such as intensive care unit (ICU) admission or death was not significantly different before and after the emergence of *delta*, although non-statistically significant increases (perhaps due to very small sample sizes) in the proportion of patients who died or required invasive mechanical ventilation were noted [9]. Likewise, the proportion of adults hospitalized with COVID-19 who died or were admitted to an ICU did not significantly change in the wake of *delta*'s emergence as the dominant strain, but the proportion of adults hospitalized with COVID who were aged 18–49 years rose, and the proportion of adults 50 and over hospitalized with COVID who experienced ICU admission or death trended upward (but was not significantly greater) [10]. Furthermore, beyond evidence surrounding *alpha* and *delta*, it is worth noting that not all increases in transmission have been tied to increases in virulence. In the case of the *D614G* mutation, clinical data do not indicate any change in the severity of disease [29]. See Alizon *et al*. [30] and Otto *et al*. [31] for a more detailed discussion of variants and how their traits relate to virulence evolution.

Clear evidence of past evolutionary increases in SARS-CoV-2 virulence suggests that future increases may be possible. Given this possibility, could either biological limits to transmission or a virulence–transmission trade-off constrain how virulent SARS-CoV-2 could become? To date, no limits to the evolution of SARS-CoV-2 transmission rate have been described. However, despite extremely large viral population sizes, only a few evolutionary increases in transmission have been detected, although this pattern should not necessarily be viewed as indicative of future trends, given the many surprises SARS-CoV-2 evolution has produced to date. Turning to the potential for a virulence–transmission trade-off to limit the evolution of virulence, several threads of epidemiological and clinical evidence indicate that disease-associated mortality does not significantly decrease transmission time in SARS-CoV-2 infection [30]. The infectious period for COVID-19 infection is known to start before symptom onset [32], the majority of infected individuals experience only mild symptoms [33], and death usually occurs several weeks after initial infection [34]. These patterns do not preclude the existence of a virulence–transmission trade-off, but they do suggest that if such a trade-off does exist, virulence likely limits transmission time through a mechanism other than death. This could occur if increased virulence is associated with increased symptom frequency (i.e. a lower asymptomatic rate, shown not to be the case for *alpha* [35]), more severe initial symptoms for those who are symptomatic, or faster symptom onset. Due to widespread public knowledge of SARS-CoV-2 circulation and non-pharmaceutical interventions such as case isolation (which has been shown to decrease transmission time as evidenced by a shortening of the serial interval [36]), individuals infected with a more virulent variant would presumably more frequently and/or more rapidly modify their behaviour to reduce onwards transmission than individuals infected with a less virulent strain. Overall, while a virulence–transmission trade-off could emerge from the relationship between transmission rate and mortality rate, or these potential links between severity, symptoms and transmission time, current data are insufficient to conclusively determine the existence of such a trade-off, its shape, or where the current SARS-CoV-2 virulence and transmission rates fall on the curve or within the set of strategies that it bounds. It is also important to note that many factors other than a virulence–transmission trade-off may influence the evolution of epidemiologically relevant SARS-CoV-2 traits, including virulence [30,31,37].

## 4. COVID-19 vaccines and virulence evolution

Current evidence indicates that the evolution of increased SARS-CoV-2 virulence is possible, as transmission and virulence may be linked, but the constraints determining the extent to which virulence might increase are not well resolved. Considering this prospect is nevertheless important, because vaccinal immunity can drive selection for increased virulence in certain circumstances. If the evolution of transmission is unconstrained, then selection always favours increased transmission (and thus increased virulence if the two are associated) regardless of the effects of immunity, which can only modify the strength of selection. However, if virulence evolution is constrained by a virulence–transmission trade-off, vaccines might have the potential to determine the strength and directionality of selection for increased virulence by removing or

reducing the costs of virulence to the pathogen. In this case, theory predicts that immunity which reduces disease but not transmission [38], or reduces disease to a greater extent than transmission [39], can drive the evolution of increased virulence. This points to a potential risk for certain vaccines to drive the evolution of more virulent pathogens. While there is evidence for some aspects of vaccine-driven pathogen evolution in several human diseases, such as the emergence of vaccine-resistant lineages of Streptoccus pneumoniae [40] and Bordetella pertussis [41], to date there have been no increases in virulence definitively linked to vaccine use in humans [42]. However, several pieces of evidence highlight the existence of this risk: evolutionary increases in virulence driven by vaccination have been observed in a mouse model of malaria [43] and in an oncogenic herpes virus infecting commercially raised chickens [44]. It is important to note that some factors present in these two examples of vaccine-driven virulence evolution are absent in the case of SARS-CoV-2. For instance, in the latter example, viral virulence was extremely high (60–100% infection fatality rate) prior to vaccine-driven evolution, and vaccinal protection against infection was extremely poor.

Vaccinal immunity to SARS-CoV-2 could open the door for virulence evolution if vaccination reduces disease to a greater extent than transmission. The contrasting consequences of SARS-CoV-2 infection in the upper respiratory tract (URT) and lower respiratory tract (LRT) provide one potential mechanism whereby the effects of immunity on these factors could be separated, and we use this to frame discussion of how such separation might drive virulence evolution. Evidence to date indicates that protection in the LRT could reduce disease severity, potentially reducing the costs of increased transmission. Protection in the URT, the primary location of infection colonization, might lead to a reduction in infection risk, potentially even to the level of sterilizing immunity, negating any gains in transmission rate the virus might be able to acquire via increased virulence [45]. The partitioning of virulence and transmission effects between respiratory tract compartments is known to exist for other respiratory diseases such as influenza [46,47]. In SARS-CoV-2, the mechanisms underlying the separation of virulence and transmission effects between the LRT and URT are increasingly resolved. The pattern of higher viral infectivity in the URT compared to the LRT reflects a decreasing gradient of angiotensin-converting enzyme 2 (the receptor protein used by SARS-CoV-2 for cellular entry) expression from the URT to the LRT [48].

Given this structure, if vaccines have differential effects in the LRT and URT, then they may have differential effects on transmission and virulence, opening the way for them to drive virulence evolution. Preliminary evidence suggests that the protective effects of SARS-CoV-2 vaccines might indeed differ between respiratory compartments. Non-human primate challenge studies investigating the efficacy of early COVID-19 vaccine candidates found that vaccinal immunity reduced viral replication in the LRT to a greater extent than in the URT [45,49]. The patterns of lower URT than LRT protection for these candidate vaccines may be rooted in the type of immune responses they induce. All are delivered intramuscularly, which generally and predominantly stimulates the production of IgG antibodies. The LRT system is primarily protected by these IgG antibodies, while the URT is primarily protected by IgA antibodies involved in mucosal immunity [45]. Other tissues where SARS-CoV-2 infection can cause acute injury [50] are protected by IgG antibodies, indicating immunological protection against damage in the lower respiratory and the rest of the body might be correlated.

While the scale of vaccine distribution indicates that vaccines will be a critical factor shaping the landscape of immunity that drives SARS-CoV-2 evolution in many parts of the world, naturally acquired immunity could also contribute to selective pressures. Evidence suggests that similarly to SARS-CoV-2 vaccines, natural immunity provides greater LRT protection than URT protection, although the differential might not be as large. In macaques re-challenged with SARS-CoV-2, immunity eliminated or significantly reduced viral replication in the LRT, and reduced replication in the URT, but to a lesser extent [51]. Consistent with this pattern, a longitudinal study of unvaccinated healthcare workers in the UK found that 0/1246 individuals with anti-spike protein IgG antibodies (from previous infection) and 89/11052 seronegative individuals became symptomatically infected, indicating that natural immunity provides robust protection against disease [52]. The same study also found that natural immunity provides substantial but incomplete immunity against infection, as the incidence of asymptomatic infection was observed to be roughly four times higher in the seronegative group compared to the seropositive group [52]. Natural infection stimulates the production of both IgG and IgA antibodies [45], which suggests that natural infection could generate more balanced URT and LRT protection than vaccination, making the evolution of increased virulence a less likely outcome of selection imposed by natural immunity alone.

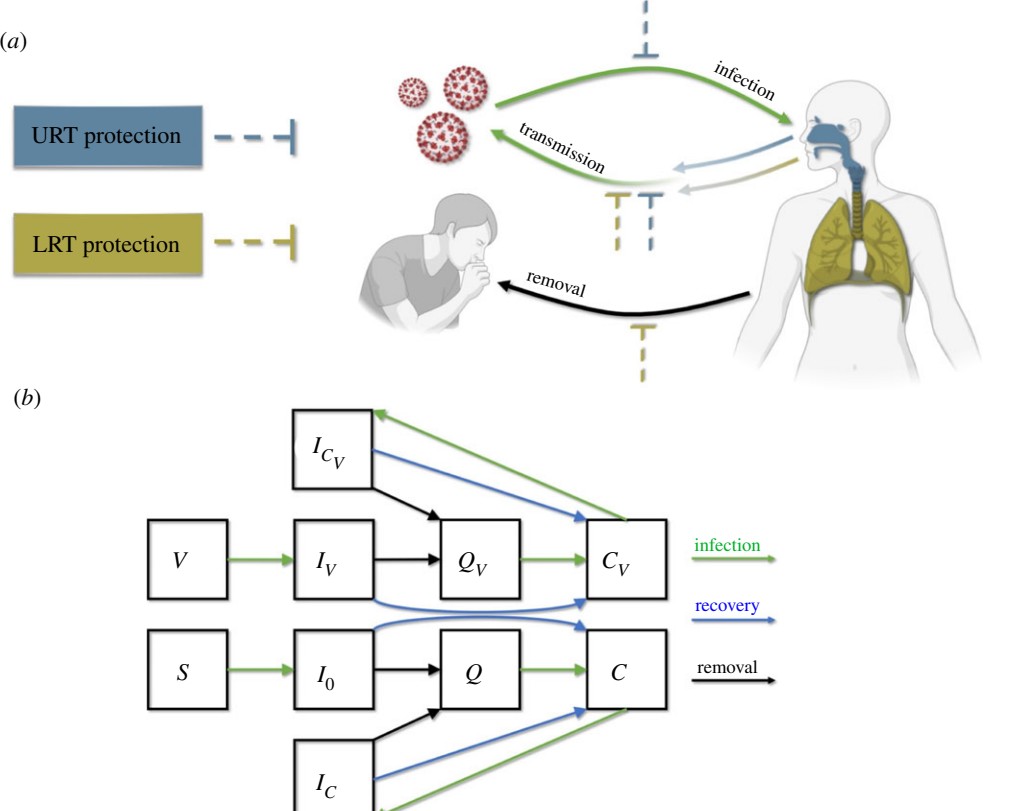

**Figure 2.** Model schematic. (*a*) The effects of immunological protection (either vaccinal or naturally acquired) in the URT and LRT on transmission, infection and disease severity. Immunological protection in the URT and/or LRT decreases the rate of onward transmission. Additionally, URT protection decreases the rate of infection, and LRT protection decreases virulence, which we assume to be proportional to the rate of removal from the transmission pool. (*b*) Structure of the epidemiological model. Susceptible ($S$) and vaccinated individuals ($V$) become infected (to classes $I_0$ and $I_V$, respectively) in a density dependent manner. Infected individuals can be removed to a quarantined class ($Q$ or $Q_V$). We assume that quarantine occurs before death in fatal infections, and as such ignore disease-associated mortality. Infected and quarantined individuals recover to a convalescent class with either naturally acquired immunity ($C$) or a combination of natural and vaccinal immunity ($C_V$). Convalescent individuals can become reinfected and return to an infected class ($I_C$ or $I_{C_V}$). All infected classes contribute to transmission.

## 5. Bounding the effects of COVID-19 vaccines on selection for increased virulence

The unprecedented scale and speed at which SARS-CoV-2 vaccines are being deployed point to the considerable value in bounding expectations on selection for increased viral virulence under different vaccine characteristics. The uncertainty surrounding the plausibility of and limits to SARS-CoV-2 virulence evolution complicates predictions, but it is possible to identify conditions associated with a pessimistic scenario in which vaccination has the most potential to drive selection for increased virulence. These conditions include the heritability of and thus evolutionary potential for increased transmission, a positive association between transmission and virulence, no biological limits to transmission, diminishing gains in transmission rate with increasing virulence, and increases in the removal rate of infectious individuals proportional to increases in virulence. These last two conditions constitute a virulence–transmission trade-off, and while they intentionally do not represent a specific mechanism, they are broadly consistent with a variety of ways in which a trade-off might emerge (e.g. faster symptom onset, more frequent isolation of symptomatic individuals). A scenario in which the relationship between transmission in virulence is positive but non-saturating would be even more pessimistic, but in this case selection for increased virulence would occur regardless of the effects of vaccination.

We develop a general theoretical framework to explore the degree to which different combinations of vaccine coverage and vaccinal protection in the URT and LRT (represented by parameters $r_{U,V}$ and $r_{L,V}$) might shape the evolution of virulence under the conditions associated with this 'pessimistic' scenario (figure 2). We take an adaptive dynamics approach to identify how vaccine characteristics map to

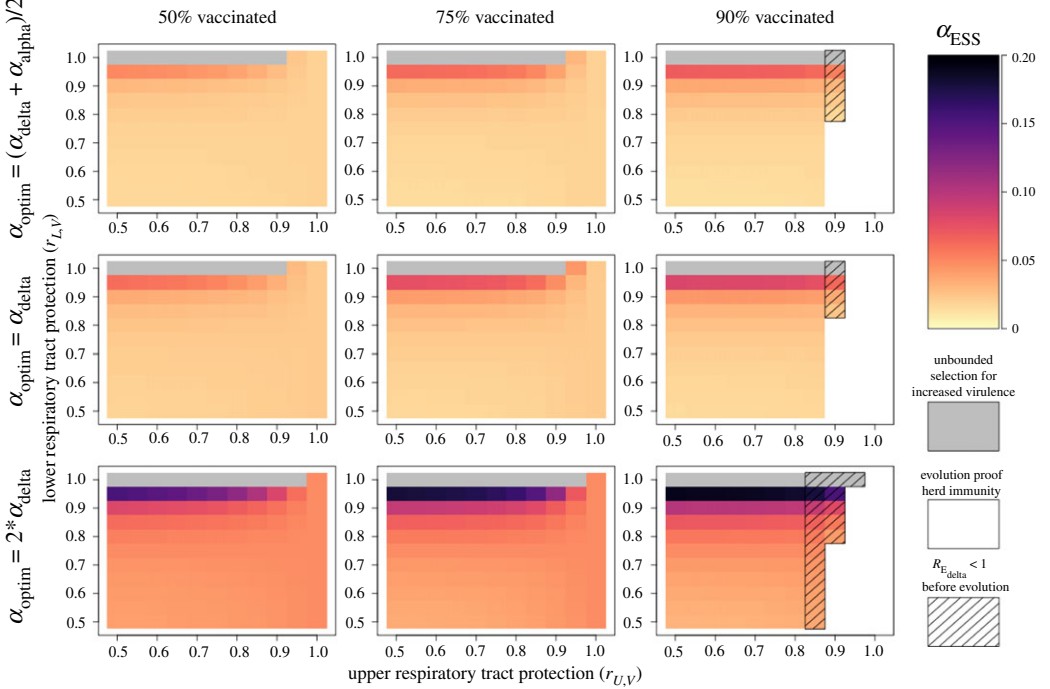

**Figure 3.** Virulence evolution and evo-epidemiological outcomes associated with vaccinal immunity. Panels show the evolutionarily stable virulence strategy resulting from different combinations of vaccinal protection in the LRT and URT. Each column corresponds to a different level of vaccine coverage, and each row corresponds to a different assumption about how the value of $\alpha_{\text{optim}}$ compares to $\alpha_{\text{delta}}$ and $\alpha_{\text{alpha}}$. Darker colours indicate more virulent evolutionary stable strategies. Grey regions correspond to scenarios in which no evolutionary stable virulence strategy exists and increased virulence is selected for without bounds. White regions correspond to scenarios in which vaccinal (and natural) immunity are sufficient to achieve herd immunity (within the theoretical population in consideration) without the possibility of virulence evolution leading to the persistence of the pathogen. Hashed lines indicate that vaccinal and natural immunity may lead to herd immunity before virulence evolution takes place, but that increases in virulence are selected for and can erode herd immunity. In all panels natural immunity effects are set to $r_{U,C} = 0.5$, $r_{L,C} = 0.75$ and the lower and upper respiratory tracts contribute equally to transmission ($\varepsilon = 0.5$).

different values of evolutionarily stable virulence (i.e. pathogen virulence at evolutionary equilibrium) and various evo-epidemiological outcomes including unbounded selection for increased virulence, evolution-proof herd immunity and non-evolution-proof herd immunity. To accommodate our mechanism-free assumption of a virulence–transmission trade-off, we assume susceptible–infectious–quarantined–convalescent dynamics. The rate at which infected individuals become quarantined is proportional to virulence, reflecting the assumption that increased virulence would lead to faster symptom onset and/or increased symptom severity. While our model formulation does not explicitly incorporate all details of COVID-19 epidemiology (e.g. pre-symptomatic transmission), its general form broadly captures all relevant dynamics. To explore how the strength and direction of selection for increased virulence changes with assumptions about where the virulence of *delta* falls on the virulence–transmission trade-off curve, we replicate our analyses across a range of assumptions about how optimal virulence, $\alpha_{\text{optim}}$, compares to the levels of virulence associated with *alpha* ($\alpha_{\text{alpha}}$) and *delta* ($\alpha_{\text{delta}}$), which is the most-fit variant detected to date. Specifically, we investigate scenarios in which $\alpha_{\text{optim}}$ is intermediate between $\alpha_{\text{delta}}$ and $\alpha_{\text{alpha}}$, equal to $\alpha_{\text{delta}}$ or greater than $\alpha_{\text{delta}}$. While *delta* has yet to be definitively linked to an increase in virulence, we assume $\alpha_{\text{alpha}} < \alpha_{\text{delta}}$, in line with our aim to investigate a pessimistic scenario. Our analysis is motivated by SARS-CoV-2, but the separation of contributions to virulence and transmission from different respiratory compartments (e.g. URT and LRT) is likely relevant for other viruses.

Our analyses reveal several important relationships between vaccinal immunity and the degree of evolutionarily stable virulence, $\alpha_{\text{ESS}}$ (figure 3). We observe that $\alpha_{\text{ESS}}$ is highly sensitive to vaccinal protection in the LRT and increases with rising values of $r_{L,V}$. This same pattern does not hold for URT protection, as $\alpha_{\text{ESS}}$ is minimally sensitive to the value of $r_{U,V}$. For any given combination of vaccine effects ($r_{U,V}$, $r_{L,V}$), $\alpha_{\text{ESS}}$ rises with the value of $\alpha_{\text{optim}}$, and to a lesser extent also increases with vaccine coverage.

Beyond determining the value of $\alpha_{ESS}$, our analyses connect vaccinal immunity to qualitative evo-epidemiological outcomes, capturing the larger overall benefits of vaccines (figure 3). Our results map to four outcomes, listed here from worst to best: endemic SARS-COV-2 with unbounded selection for increased virulence (figure 3, grey regions), endemic SARS-COV-2 with an increase in virulence to $\alpha_{ESS}$ (figure 3, orange to purple regions), herd immunity (within the theoretical population in consideration) if epidemiological equilibrium is reached before an evolutionary increase in virulence to $\alpha_{ESS}$ occurs (figure 3, hashed regions), and evolution-proof herd immunity (figure 3, white regions). Across all assumptions about the optimum degree of virulence, the latter two outcomes can only be achieved when URT protection is strong and vaccine coverage is high, with the ideal outcome of evolution-proof herd immunity requiring higher levels of URT protection. These positive outcomes were achievable across all assumed values of $\alpha_{optim}$ for similar degrees of URT and LRT protection. We observed similar patterns of virulence evolution and evo-epidemiological outcomes when assuming that LRT infection contributes minimally to transmission and for alternative assumptions about the strength of the protective effects of natural immunity in the URT and LRT (electronic supplementary material, figures S3–S5).

Clinical trials and real world vaccine efficacy data provide information about the effects of COVID-19 vaccines that we can compare to our model predictions to shed light on how we might expect virulence evolution to unfold. Evidence from early clinical trials suggested that COVID-19 vaccines provide robust protection against disease. Two mRNA vaccines, BNT162b2 and mRNA-1723, were shown to be 95% and 94% effective in preventing COVID-19, respectively, while also exhibiting promise of preventing severe symptoms [3,4]. A recombinant adenovirus vaccine, ChAdOx1, had an efficacy of 62–90% in clinical trials carried out in Brazil, South Africa and the UK [5]. Another recombinant adenovirus vaccine, Gam-COVID-Vac or Sputnik V, exhibited an efficacy of 92% [6]. This strong protection against severe disease appears to hold against the *delta* variant as evidenced by real-world data. In a large test-negative study in Canada, mRNA-1723, BnT162b2 and ChAdox1 were shown to be 98%, 97% and 91% effective against the *delta* variant in preventing hospitalization and death after two doses [53].

Clinical trials provided more limited information about the infection-limiting effects of vaccines, but the results were nonetheless promising. The ChAdOx1 vaccine was shown to reduce the incidence of PCR confirmed infection by 67% [54], consistent with a moderately strong infection blocking effect. An even more striking result was observed for the BNT162b2 vaccine, which has been shown to be 95.3% effective in preventing infections [55]. Real-world data indicate that vaccines may be less effective against the *delta* variant in preventing transmission versus severe disease. The test-negative study in Canada found that after two doses, mRNA-1723, BNT162b2 and ChAdOx1 were 95%, 92% and 87% effective against *delta* in preventing symptomatic infection, respectively [53]. Data from England indicate slightly lower efficacy in protection against symptomatic infection, with two doses of BNT162b2 and ChAdOx1 estimated to be 88% and 67%, respectively [56]. Concerningly, the efficacy of vaccination against infection appears to wane significantly over time [57,58]. Booster shots have been shown to be effective in restoring robust protection at least temporarily [59], and could provide a means for continued vaccinal protection against infection in the long term. Data on the extent to which SARS-COV-2 vaccines reduce onwards transmission are currently limited.

As a whole, this evidence suggests that COVID-19 vaccines confer a very high degree of LRT protection, and likely a significant amount of URT protection, although it is not possible at this time to map vaccines to a specific combination of protective effects ($r_{U,V}$, $r_{L,V}$), and the emergence of vaccine-resistant strains may decrease vaccine efficacy [60]. Given this combination of effects, our model suggests that (i) patterns of virulence evolution are likely to be highly sensitive to protective effects of vaccination and the optimum degree of virulence, and thus are impossible to predict at this time, (ii) the optimal level of virulence, $\alpha_{ESS}$, is expected to increase slightly with vaccine coverage, (iii) existing vaccines likely significantly reduce transmission and could, if administered to a sufficient proportion of a population, generate herd immunity even under conditions where virulence evolution could occur (noting that waning vaccinal protection could jeopardize this outcome), and (iv) these positive outcomes would remain attainable even if $\alpha_{optim}$ is somewhat higher than $\alpha_{delta}$.

# 6. Discussion

As the global spread of SARS-CoV-2 continues and the widespread rollout of vaccines is underway, evaluating the risk of vaccine-driven virulence evolution increases in priority. We reviewed the

theoretical drivers of virulence evolution, distilled a set of conditions necessary for the evolution of increased virulence, and evaluated them in the context of SARS-CoV-2, concluding that further evolution of SARS-CoV-2 virulence is plausible, but not certain. While the limits to SARS-CoV-2 evolution are not clear, several mechanisms could potentially result in vaccines driving selection for or against increased virulence. Using an evo-epidemiological model, we determined that under a pessimistic set of assumptions, existing vaccines could potentially select for increased virulence, but that if this were to occur, they would still generate herd immunity if administered to a large percentage of the population. This work builds upon the theoretical analyses of SARS-CoV-2 virulence evolution presented in many previous studies [30,37] by extending the same fundamental concepts to the issue of vaccine-driven evolution.

The concept of herd immunity in the context of our results warrants careful consideration. Herd immunity does not necessarily correspond to a 'COVID-free' world [61]. Even if a population has immunological characteristics that render $R_E < 1$, spillover transmission from other populations could lead to small outbreaks of SARS-CoV-2, especially in sub-populations with low vaccination rates. However, herd immunity can prevent sustained within-population transmission. Another important detail deals with the temporal aspects of herd immunity. Although vaccines could create herd immunity in the short term, a build-up of susceptible individuals due to births and immunological waning might result in the loss of this population-level protection, leading to epidemics. While we do not explicitly incorporate waning or demographic processes into our predictions, interpreting the values of $r_{U,V}$ and $r_{L,V}$ as population means that encompass decreases in efficacy over time allows for an approximate estimate of the impacts of waning (see Methods).

The large degree of uncertainty surrounding the plausibility of vaccine-driven virulence evolution, combined with our finding that vaccines could lead to herd immunity even if they select for increased virulence, suggests no cause for immediate concern about the risks posed by vaccines regarding virulence evolution. Additionally, the burden of increased virulence could be significantly decreased if virtually all primary infections occur in childhood, resulting in mild symptoms. In the absence of virulence evolution, this pattern is predicted to decrease the severity of SARS-CoV-2 to a level similar to other endemic human coronaviruses [62]. However, several actions should be taken to mitigate any risks. First, the protective effects of all SARS-CoV-2 vaccines in the URT and LRT, and how these effects map to reductions in infection and transmission, should be evaluated, further noting that this categorization may also be a simplification, and other mechanisms may be at play and should be investigated. The use of any vaccine that does not significantly reduce contagion should be carefully considered. Second, as more vaccines and more data become available, governments should prioritize the use of vaccines (including booster doses) that provide robust protection against both disease and transmission. Third, vaccination programmes should aim to immunize a large percentage of the population as quickly as possible, as herd immunity, which could mitigate selection for increased virulence, is only possible at high vaccine coverage (in the absence of non-pharmaceutical interventions).

The benefits of transmission/infection reducing immunity stemming from vaccinal URT protection extend beyond minimizing selection for increased virulence and generating herd immunity. These effects, along with other measures, such as non-pharmaceutical interventions, would also decrease effective viral population size, diminishing the impact of selection, the genetic diversity of the virus, and thus the likelihood of vaccine-driven virulence evolution [37]. However, sources of genetic diversity could exacerbate risks of viral evolution. Significant viral diversity can be generated in chronically infected immunocompromised patients, and enhanced infection control procedures have been recommended to limit the spread of mutations from these individuals. Animal reservoirs are another potential source of genetic diversity [63]. SARS-CoV-2 epidemics in mink farms have already been linked to the generation of novel mutations [64] and animal to human transmission [65]. Other domestic and agricultural animal populations, especially those maintained at high density, should be actively monitored for SARS-CoV-2 transmission. Wild animal populations also have potential to harbour sustained viral transmission but present significant challenges for implementing control measures. Species that are known to be competent hosts for coronaviruses and regularly come into contact with humans should be prioritized for monitoring.

If vaccines were to drive the evolution of increased virulence, protection against disease in vaccinated individuals may or may not erode, but other groups would bear greater impacts [39]. Specifically, unvaccinated individuals would bear the consequences of increased virulence without any vaccinal protection to offset disease severity. Vaccine-driven evolution could also amplify healthcare disparities between populations if a SARS-CoV-2 strain with increased virulence evolves in a population with a

high vaccination rate, and spills over into a population with a low vaccination rate. Beyond seeking to reduce disparities in vaccination, countries with the highest vaccine coverage should continue to engage in virulence monitoring to detect emergent variants with increased virulence, and if such variants arise, enact containment measures to mitigate negative consequences for countries with limited access to vaccines.

Immunological waning is only implicitly incorporated in our analyses, but it is important to note that the resulting loss of vaccinal and/or natural immunity over time could increase both the effective viral population sizes and likelihood of SARS-CoV-2 becoming endemic [62,66,67], while also altering the selective forces acting on pathogen virulence. Future analyses should explore how waning immunity might alter selection for increased virulence as evidence regarding the longevity of immunity emerges. Booster vaccine doses offer a potential solution to mitigate waning vaccinal immunity. If waning vaccinal immunity results in a decrease in protection against infection but does not reduce protection against disease, then booster vaccine doses should be evaluated primarily on their ability to prevent infection and transmission.

While our analyses give approximate expectations under a specific set of assumptions, robust expectations about the likelihood, magnitude and trajectory of virulence evolution remain elusive. We assumed a conventional saturating positive relationship between transmission and virulence. However, the shape, nature and existence of any trade-offs governing virulence evolution in SARS-CoV-2 are currently unknown, and the form of true relationship could modify the direction and magnitude of selection acting on virulence. Assessments of the risk of virulence evolution should be updated as new findings emerge, both by directly monitoring the virulence of the virus, and by better characterizing the nature of the trade-off. To monitor virulence, infection fatality ratios associated with circulating SARS-CoV-2 strains should continue to be surveilled to detect early evidence of any evolutionary trends. Stratifying these observations by age and immunological status (naive, natural immunity, vaccinal immunity) would allow for changes in population-wide infection fatality rates to be properly attributed changes in virulence or other factors. Decreases in fatality rates tied to medical advances will complicate this monitoring, especially as they vary globally, but animal or cell-culture models might provide a means for standardized measurement. Trade-offs affecting the evolution of SARS-CoV-2 transmission can be characterized through comparative epidemiological analyses and clinical studies of within-host viral dynamics.

While we focus on the evolution of SARS-CoV-2 virulence and transmission in this paper, other aspects of viral evolution may also affect COVID-19 severity. For example, antigenic evolution may have implications for virulence. This could occur either through direct effects, such as antigenic changes altering the function of proteins involved in transmission or pathogenesis, or through indirect effects, such as antibody-dependent enhancement [68], which could lead to increases in viral load and ultimately greater disease severity.

Vaccines have the ability to provide protection against COVID-19 disease at a massive scale, but widespread protection could impose significant selection pressures on the SARS-CoV-2 virus. Current evidence suggests no immediate cause for concern about virulence evolution, but careful monitoring and efficacy evaluation are needed moving forward.

# 7. Methods

## 7.1. Epidemiological model

To evaluate the potential evolutionary consequences associated with SARS-CoV-2 vaccine candidates, we develop a compartmental ordinary differential equation model that broadly reflects SARS-CoV-2 epidemiology and the protective effects of naturally acquired and vaccine-induced immunity (figure 2). We separate the impacts of vaccinal and natural immunity in the upper and lower respiratory tracts, and the impact of these protective effects on infection, transmission and virulence. We assume that infection in the LRT and URT additively contributes to transmission rate, $\beta$ (equation (7.1)). The parameter $\varepsilon$ defines the fractional contribution of the LRT to transmission. Transmission rates in both the URT and LRT follow an increasing and saturating function of virulence defined by shape parameters $b_1$ and $b_2$. We assume that a virulence rate greater than 0.0025 is necessary for transmission (equation (7.1)). Immunity limits transmission by reducing virulence by a scalar $(1 - r_{U,V})$ in the URT, and $(1 - r_{L,V})$ in the LRT (in this case transmission is calculated as $\beta(\alpha,\ r_{U,V},\ r_{L,V})$). Naturally acquired immunity and a combination of naturally acquired

and vaccinal immunity limit transmission in the same ways according to the parameters $r_{U,C}$, $r_{L,C}$ and $r_{U,C_V}$, $r_{L,C_V}$ respectively.

$$\beta(\alpha, r_U, r_L) = (1 - \varepsilon)b_1((1 - r_U)(\alpha - 0.0025))^{b_2} + \varepsilon\, b_1((1 - r_L)(\alpha - 0.0025))^{b_2}. \qquad (7.1)$$

Moving from how immunity affects transmission to considering how it might affect infection and virulence, we begin by connecting virulence to the rate of removal of infectious individuals from the transmission pool. We set this rate equal to virulence ($\alpha$) times a constant ($p$). This association between virulence and the removal of infectious individuals combined with the form of the function relating virulence to transmission constitutes a virulence–transmission trade-off (figure 1). To incorporate the effects of immunity on virulence, we assume that only LRT protection reduces virulence, in line with current reporting on the pathology of SARS-CoV-2. The effects of virulence on disease-associated removal are reduced by a scalar $(1 - r_{L,V})$ for vaccinated individuals (in class $V$), by $(1 - r_{L,C})$ for convalescent individuals with naturally acquired immunity (in class $C$), and by $(1 - r_{L,C_V})$ for individuals with naturally acquired and vaccinal immunity (in class $C_V$). To model effects of immunity on transmission, we assume that the force of infection ($\lambda$) is reduced by a scalar $(1 - r_{U,V})$ for individuals in class $V$, by $(1 - r_{U,C})$ for individuals in class $C$, and by $(1 - r_{U,C_V})$ for individuals in class $C_V$ as the URT seems to be the key driver of transmission. Note that convalescent individuals (in both the $C$ and $C_V$ classes) can become reinfected if immunity provides incomplete protection against reinfection ($r_{U,C}$ or $r_{U,C_V} < 1$).

This framework for modelling the effects of immunity is designed to capture key differences between URT and LRT protection. To consider how vaccinal and natural immunity affect infection, transmission and virulence without separating the effects of immunity between respiratory compartments, this framework would need to be adjusted to decouple reductions in transmission from reductions in infection and virulence.

After defining the relationships between immunity, transmission, infection and virulence, we incorporate them into a compartmental epidemiological model (figure 2$b$; equations (7.1)–(7.3)). We assume a 1 day time step, and accordingly set the recovery rate $\gamma = 1/7$ [69]. We ignore disease-associated mortality, and assume that isolation occurs before death. This is consistent with the observation that death usually occurs several weeks after the appearance of symptoms. We assume that after individuals are isolated, they enter a quarantine class ($Q_V$ for those individuals who did have vaccinal immunity prior to infection, and $Q$ for individuals who did not) in which they can neither contribute to transmission nor be reinfected. Individuals move from quarantined classes to convalescent classes (either $C$ or $C_V$) at rate $q = 1/10$ as they recover from infection and no longer contribute to transmission. Immunity wanes at rate $\omega$ for the convalescent class ($C$), and at rate $\omega_V$ for the vaccinated ($V$) and vaccinated-convalescent class ($C_V$). We ignore birth and death processes to hold the population size constant.

$$\lambda = \beta(\alpha, 0, 0)\, I_0 + \beta(\alpha, r_{U,V}, r_{L,V})\, I_V + \beta(\alpha, r_{U,C}, r_{L,C})\, I_C + \beta(\alpha, r_{U,C_V}, r_{L,C_V})\, I_{C_V}, \qquad (7.2)$$

$$\frac{dS}{dt} = -S\,(\lambda + \mu) + C\,\omega + C_V \omega_V, \qquad (7.3a)$$

$$\frac{dV}{dt} = -V((1 - r_{U,V})\,\lambda), \qquad (7.3b)$$

$$\frac{dI_0}{dt} = S\,\lambda - I_0\,(\gamma + \alpha\,p), \qquad (7.3c)$$

$$\frac{dI_V}{dt} = V\,(1 - r_{U,V})\,\lambda - I_V\,(\gamma + (1 - r_{L,V})\,\alpha\,p), \qquad (7.3d)$$

$$\frac{dI_C}{dt} = C\,(1 - r_{U,C})\,\lambda - I_C\,(\gamma + (1 - r_{L,C})\,\alpha\,p), \qquad (7.3e)$$

$$\frac{dI_{C_V}}{dt} = C_V\,(1 - r_{U,C_V})\,\lambda - I_{C_V}\,(\gamma + (1 - r_{L,C_V})\,\alpha\,p), \qquad (7.3f)$$

$$\frac{dQ}{dt} = I_0\,\alpha\,p + I_C\,(1 - r_{L,C})\,\alpha\,p - Q\,q, \qquad (7.3g)$$

$$\frac{dQ_V}{dt} = I_V(1 - r_{L,V})\,\alpha\,p + I_{C_V}\,(1 - r_{L,C_V})\,\alpha\,p - Q_V q, \qquad (7.3h)$$

$$\frac{dC}{dt} = I_0\,\gamma + I_C\,\gamma + Q\,q - C\,((1 - r_{U,C})\,\lambda + \omega) \qquad (7.3i)$$

and

$$\frac{dC_V}{dt} = I_V\,\gamma + I_{C_V}\,\gamma + Q_V q - C_V\,((1 - r_{U,C_V})\,\lambda + \omega_V). \qquad (7.3j)$$

## 7.2. Evolutionary analysis

The dynamics of the above model hinge on the relationship between transmission and virulence, and as such are sensitive to the shape of the virulence–transmission trade-off. We aim to explore how the epidemiological and evolutionary outcomes predicted by this model change with assumptions about the effects of vaccinal immunity and where the virulence of the *delta* variant falls on the trade-off curve. We begin by defining the observed values of virulence associated with the ancestral strain and evolved variants. Building from an estimated infection fatality rate of the ancestral strain ($\alpha_{\mathrm{ansc}}$) of 0.005 [70], we set $\alpha_{\mathrm{alpha}} = 1.5^*\alpha_{\mathrm{ansc}} = 0.0075$. We assume the *delta* variant to be twice as virulent as the ancestral strain and set $\alpha_{\mathrm{delta}} = 2^*\alpha_{\mathrm{ansc}} = 0.01$.

Next, we define values of $R_0$ across COVID-19 strains. The transmission rate of *delta* is estimated to be approximately 1.5 times greater than that of the *alpha* variant [71], which in turn was estimated to be approximately 1.5 times greater than that of the ancestral strain. Given the basic reproduction number of the ancestral strain $R_{0_{\mathrm{ansc}}} = 2.5$ [72–74], we set $R_{0_{\mathrm{alpha}}} = 1.5 * R_{0_{\mathrm{ansc}}} = 3.75$ and $R_{0_{\mathrm{delta}}} = 1.5 * R_{0_{\mathrm{alpha}}} = 5.625$. Following this, we set $p$, the constant relating virulence to the rate of removal, to 50 so that the average amount of time an individual (infected with the *delta* strain) is infectious before isolation is $1/\alpha_{\mathrm{delta}} * p = 2$ days if recovery is ignored. This value is roughly consistent with the assumptions of other models [75].

Following this, we construct a set of optimal virulence ($\alpha_{\mathrm{optim}}$) values to explore that reflects various assumptions about the fitness of $\alpha_{\mathrm{delta}}$ relative to the optimum. Because the *delta* strain is assumed to be more fit than the *alpha* strain (i.e. it has a higher $R_0$), and the *alpha* strain more fit than the ancestral strain, we assume that the optimum virulence is either intermediate between $\alpha_{\mathrm{alpha}}$ and $\alpha_{\mathrm{delta}}$, equal to $\alpha_{\mathrm{delta}}$ or greater than $\alpha_{\mathrm{delta}}$: $\alpha_{\mathrm{optim}} \in \{\alpha_{\mathrm{alpha}} + \alpha_{\mathrm{delta}}/2, \alpha_{\mathrm{delta}}, 2 * \alpha_{\mathrm{delta}}\}$. Electronic supplementary material, figure S1, illustrates how this set of assumptions corresponds to different sets of relationships between virulence, transmission, transmission time and fitness.

We use the next-generation matrix approach [76] to derive the following equation for the effective reproductive number $R$:

$$F = \begin{bmatrix} S*\beta(\alpha,0,0) & S*\beta(\alpha,r_{U,V},r_{L,V}) & S*\beta(\alpha,r_{U,C},r_{L,C}) \\ V*(1-r_{U,V})*\beta(\alpha,0,0) & V*(1-r_{U,V})*\beta(\alpha,r_{U,V},r_{L,V}) & V*(1-r_{U,V})*\beta(\alpha,r_{U,C},r_{L,C}) \\ C*(1-r_{U,C})*\beta(\alpha,0,0) & C*(1-r_{U,C})*\beta(\alpha,r_{U,V},r_{L,V}) & C*(1-r_{U,C})*\beta(\alpha,r_{U,C},r_{L,C}) \\ C_V*(1-r_{U,C_V})*\beta(\alpha,0,0) & C_V*(1-r_{U,C_V})*\beta(\alpha,r_{U,V},r_{L,V}) & C_V*(1-r_{U,C_V})*\beta(\alpha,r_{U,C},r_{L,C}) \\ 0 & 0 & 0 \\ 0 & 0 & 0 \end{bmatrix}$$

$$\begin{bmatrix} S*\beta(\alpha,r_{U,C_V},r_{L,C_V}) & 0 & 0 \\ V*(1-r_{U,V})*\beta(\alpha,r_{U,C_V},r_{L,C_V}) & 0 & 0 \\ V*(1-r_{U,V})*\beta(\alpha,r_{U,C_V},r_{L,C_V}) & 0 & 0 \\ C_V*(1-r_{U,C_V})*\beta(\alpha,r_{U,C_V},r_{L,C_V}) & 0 & 0 \\ 0 & 0 & 0 \\ 0 & 0 & 0 \end{bmatrix} \quad (7.4a)$$

$$V = \begin{bmatrix} \gamma + \alpha\,p & 0 & 0 & 0 & 0 & 0 \\ 0 & \gamma + (1-r_{L,V})\,\alpha\,p & 0 & 0 & 0 & 0 \\ 0 & 0 & \gamma + (1-r_{L,C})\,\alpha\,p & 0 & 0 & 0 \\ 0 & 0 & 0 & \gamma + (1-r_{L,C_V})\,\alpha\,p & 0 & 0 \\ -\alpha\,p & 0 & -(1-r_{L,C})\,\alpha\,p & 0 & q & 0 \\ 0 & -(1-r_{L,V})\,\alpha\,p & 0 & -(1-r_{L,C_V})\,\alpha\,p & 0 & q \end{bmatrix} \quad (7.4b)$$

and

$$R = \text{The spectral radius of } F*V^{-1}. \quad (7.4c)$$

When evaluated for a completely susceptible population, equation (7.4) gives $R_0$, the basic reproductive number. For a given assumption about the value of $\alpha_{\mathrm{optim}}$, we used equation (7.4) and numerical methods to set the value of $b_2$ to that which maximizes $R_0$ at $\alpha_{\mathrm{optim}}$ and $b_1 = 1$, and the value of $b_1$ to that which corresponded $R_0 = R_{0_{\mathrm{delta}}}$ at $\alpha_{\mathrm{delta}}$. We checked to ensure that the value of $b_2$ was insensitive to the value of $b_1$, that $\alpha_{\mathrm{optim}}$ maximized $R_0$, and that $\alpha_{\mathrm{delta}}$ corresponded to $R_0 = R_{0_{\mathrm{delta}}}$ at the identified values of $b_1$ and $b_2$. Electronic supplementary material figure S1 shows the virulence transmission trade-off curves constructed from the three assumptions about the value of $\alpha_{\mathrm{optim}}$; these assumptions map to a wide range of possible shapes of the trade-off.

To characterize the outcome of pathogen virulence evolution in response to vaccination we identify how a range of vaccinal protection characteristics in the URT and LRT map to epidemiological outcomes before pathogen evolution occurs, as well as patterns of virulence after many iterative virulence evolution events. Because SARS-CoV-2 virulence evolution has been observed to play out over a matter of months, we ignore processes that operate on longer time scales such as births and deaths. Additionally, we set $\omega = 0$ and $\omega_V = 0$ so that immunity does not wane. While immunological waning certainly does occur, its effects are likely small on the time scale of virulence evolution. In the adaptive dynamics approach that we employ (see below), incremental changes in pathogen virulence are assumed to occur only after epidemiological equilibrium is reached. Including immunological waning would force the model to take many years to reach equilibrium, making our predictions fall significantly out of step with the observed rate of pathogen evolution. An alternative way to incorporate the effects of immunological waning into our analyses is to consider the values of the parameters $r_{U,V}$ and $r_{L,V}$ to reflect mean population values that account for variation in vaccinal immunity across individuals. As such, the effects of immunological waning could be conceptualized as decreasing the values of $r_{U,V}$ and $r_{L,V}$. Likewise, booster vaccine doses could be conceptualized as increasing the values of these two parameters.

We iterate our analysis for various scenarios encompassing different vaccination rates and values of $\alpha_{\text{optim}}$. We only consider values of $r_{U,V}$ and $r_{L,V}$ in [0.5, 1.0] as all widely used vaccines have been shown to be at least 50% effective in preventing infection and severe disease [77]. In the scenarios presented in the main text, we set $r_{U,C} = 0.5$ and $r_{L,C} = 0.75$. Electronic supplementary material figures S3–S5 give results for alternative scenarios in which we consider stronger ($r_{U,C} = 0.75$ and $r_{L,C} = 0.9$) and weaker ($r_{U,C} = 0.25$ and $r_{L,C} = 0.5$) effects of natural immunity. Additionally, because vaccinated individuals who experience breakthrough infections exhibit immunity stronger than either vaccinal or natural immunity alone [78], we set $r_{U,C_V} = (1 + r_{U,V})/2$ and $r_{L,C_V} = (1 + r_{L,V})/2$.

For each unique combination of $r_{U,V}$, $r_{L,V}$, $\alpha_{\text{optim}}$, and vaccination rate ($v$), we first initialize the model assuming that 37.5% of individuals are convalescent (roughly consistent with the estimated cumulative incidence in the USA as of May 2021 [79]), that 0.1% of individuals are infected, and that a fraction $v$ (with $v \leq 1 - 0.001$) are vaccinated. Using these assumptions, we set the initial conditions as follows:

$$S(0) = 1 - C_V(0) - C(0) - V(0) - 0.001, \tag{7.5a}$$

$$V(0) = (1 - 0.375)\, v, \tag{7.5b}$$

$$I(0) = 0.001\, \frac{S(0)}{C_V(0) + C(0) + V(0) + S(0)}, \tag{7.5c}$$

$$I_V(0) = 0.001\, \frac{V(0)}{C_V(0) + C(0) + V(0) + S(0)}, \tag{7.5d}$$

$$I_C(0) = 0.001\, \frac{C(0)}{C_V(0) + C(0) + V(0) + S(0)}, \tag{7.5e}$$

$$I_{CV}(0) = 0.001\, \frac{C_V(0)}{C_V(0) + C(0) + V(0) + S(0)}, \tag{7.5f}$$

$$Q(0) = 0, \tag{7.5g}$$

$$Q_V(0) = 0, \tag{7.5h}$$

$$C(0) = 0.375\, (1 - v) \tag{7.5i}$$

and

$$C_V(0) = 0.375\, v. \tag{7.5j}$$

Next, we calculate the effective reproductive number ($R_E$) for $\alpha_{\text{delta}}$ by evaluating equation (7.4) at these starting conditions. $R_E$ values are informative about the short term epidemiological consequences of vaccination before pathogen evolution acts. Values less than one indicate that herd immunity is achieved (within the theoretical population in consideration), while values greater than or equal to one indicate that this is not the case.

After calculating $R_E$ for $\alpha_{\text{delta}}$ at the starting conditions, we proceed with an adaptive dynamics approach to identify the evolutionarily stable virulence strategy ($\alpha_{\text{ESS}}$) associated with the $r_{U,V}$, $r_{L,V}$, $\alpha_{\text{optim}}$, and $v$ parameters. Evolutionarily stable virulence strategies can resist invasion by any other strategy (i.e. $R_E < 1$ for $\alpha \neq \alpha_{\text{ESS}}$), and as a result are expected to be the equilibrium strategy over evolutionary time. We consider potential virulence strategies in the set $A = \{\{0.0025, 0.0030, 0.0035, \ldots, 0.0500\}, \{0.055, 0.06, 0.065, \ldots, 0.4000\}\}$. For each $\alpha_{\text{resident}} \in A$, we (i) set $\alpha = \alpha_{\text{resident}}$ and simulate epidemiological dynamics over 1 year using the R package 'desolve' [80] and the 'lsoda' integrator with a step size of 1 day, (ii) extract frequencies of individuals in each class as the epidemiological equilibrium conditions, and finally (iii) for each $\alpha_{\text{invader}} \in A$, set $\alpha = \alpha_{\text{invader}}$ and calculate $R_E$ by evaluating equation (7.4) at the epidemiological equilibrium conditions. We then plot the ability of each $\alpha_{\text{invader}}$ strategy to invade each $\alpha_{\text{resident}}$ strategy

(i.e. $R_E > 1$) to generate pairwise invasibility plots, and use these plots to identify evolutionarily stable strategies (electronic supplementary material, figure S2).

Data accessibility. Data and relevant code for this research work are stored in GitHub: https://github.com/ianfmiller/covid_vaccines_virulence_evolution and have been archived within the Zenodo repository: https://zenodo.org/badge/latestdoi/315507829.

Authors' contributions. I.F.M.: conceptualization, formal analysis, methodology, project administration, software, visualization, writing—original draft, writing—review and editing; C.J.E.M.: conceptualization, supervision, writing—original draft, writing—review and editing.

All authors gave final approval for publication and agreed to be held accountable for the work performed therein.

Competing interests. The authors declare no competing interests.

Funding. I.F.M. is supported by a National Science Foundation Graduate Research Fellowship.

Acknowledgements. We thank Bryan Grenfell, Jeremy Farrar, Gordon Douglas, Daniel Douek and Adrian McDermott for helpful discussions. Figure 2a was created in part using BioRender.com.

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
