## [Peer Review File · Royal Society Open Science]

Review History

RSOS-211021.R0 (Original submission)

Review form: Reviewer 1

Is the manuscript scientifically sound in its present form?

Yes

Are the interpretations and conclusions justified by the results?

Yes

Is the language acceptable?

Yes

Do you have any ethical concerns with this paper?

No

Have you any concerns about statistical analyses in this paper?

No

Recommendation?

Major revision is needed (please make suggestions in comments)

Comments to the Author(s)

I enjoyed reading Miller and Metcalf's manuscript on assessing the risk of virulence evolution in SARS-CoV-2 as a response to vaccination. It is well written overall, and shows a clear understanding of virulence evolution theory. Given the timeliness of the topic, there is a need for more theory papers addressing evolutionary trajectories of SARS-CoV-2, particularly in light of growing evidence that our most common vaccines are, in fact, more "leaky" than hoped. I enjoyed the attention to detail of the two regions of infection that are linked to transmission (URT) and severe disease (LRT) and appreciate the inclusion of this without having to overly complicate the model.

While the methods are sound overall and the model is simple enough to answer the question, the manuscript is considerably outdated in various ways. I understand covid research moves very quickly but any paper to be posted as a preprint this summer and published this fall should include more recent examples, data, and discussion of the most important VOCs currently under investigation.

There are a couple glaring omissions:

- 1) There is a lack of references to recent papers on SARS-CoV-2 evolution theory or commentaries by important evolutionary biologists in this subfield. Namely: Day et al. 2021 Current Biology, Otto et al. 2021 Current Biology, and in particular Alizon and Sofonea 2021 J. Evo.Bio. As indicated below, there are several places where these (and some of their citations within) should be cited and the authors results be discussed in the context of these works (e.g. in the discussion section).
- 2) The Delta variant needs to be included. I understand that this manuscript may have been written at a time before Delta had emerged but unfortunately, this variant is too important evolutionarily to omit. It has both increases in transmission and virulence (e.g. Fisman and Tueite 2021 medrxiv) and reductions in vaccine efficacy. The authors consider cases where the optimum virulence is either intermediate between the ancestral strain and B.1.1.7 or greater than B.1.1.7, however, given how delta is sweeping in various geographical regions, it's pretty clear now that the case where the optimum is below B.1.1.7's virulence is not likely. The question in everyone's mind is, how high can virulence get? Please run the model calibrated for the delta variant's parameters (as those done for B.1.1.7 in lines 495-516) and change figures 3 and 4 to include only B.1.1.7, delta and a hypothetical "more virulent than delta" variant, given that the $\alpha_{\text{optim}} = 1.25 \cdot \alpha_{\text{anc}}$ case is no longer relevant.

Other comments

Fig. 1: The first figure should be a schematic of the transmission-virulence trade-off, particularly showing the different shapes it might have. The section "theory of virulence evolution" is nicely written but it's confusing to use a results image as an illustrative image of the concept (e.g. the unknown parameters and how they are related is confusing given the equations/model have not been presented yet). A schematic of the trade-off could be added as the first panel to Fig.1 or have a new stand-alone figure in the introduction.

Paragraph ending in line 120: Delta needs to be added here.

Paragraph starting in line 121: please cite and discuss previous published works mentioned number 1 above.

Line 132: this decoupling of death from the trade-off has been discussed in the worked mentioned in number 1, please cite.

Paragraph starting on line 285: please include real-world data here not just the vaccine trials, since we now have estimates for these in global populations.

A direct analysis of vaccine leakiness (how much infectious viral shedding per vaccinated host) and its potential to drive virulence evolution would require a nested model approach, which is, understandably, beyond the scope of this study. However, given parallels of our current vaccine situation with that of Marek's Disease virus (which the authors briefly mention), the most studied example of vaccine-driven virulence evolution, it would be of use to the reader to understand the authors' results in context of this example (e.g. Read et al. 2015 Plos Biology). Please include this in the discussion section.

Points relating to herd immunity:

Vaccine-induced herd immunity is an important part of the results of this paper and given that reaching HIT (herd-immunity threshold) is a moving target for SARS-CoV-2 (e.g. Hodgson et al. 2021 Eurosurveillance), here are some specific points to address about herd immunity.

The underlying R_0 of a virus really affects the needed level of coverage to reach HIT and delta has changed this. How do the authors' results of their scenarios in Fig. 4, such as "how likely herd immunity without virulence evolution", change when considering the delta variant?

Line 306: "... 3) existing vaccines ..." this is still theoretically true but please cite/add any caveats that have been debated in the modelling literature. For example, the vaccines don't reduce transmission as much as we hoped (especially with 2 doses and for delta) and the projections for vaccine coverages needed for reaching herd immunity are changing (e.g. Mancuso, Eikenberry and Gumel 2021 medrxiv).

In figures 3 and 4, the 10% vaccinated coverage panels are there for completeness but are not really what we are aiming for globally. It would make more sense to just present two columns of 50 and 90 or have a gradation from 50 to 100%.

Review form: Reviewer 2

Is the manuscript scientifically sound in its present form?

No

Are the interpretations and conclusions justified by the results?

No

Is the language acceptable?

Yes

Do you have any ethical concerns with this paper?

No

Have you any concerns about statistical analyses in this paper?

No

Recommendation?

Major revision is needed (please make suggestions in comments)

Comments to the Author(s)

The authors provide a well-written overview of how SARS-CoV-2 might evolve towards greater virulence in response to numerous factors, such as vaccination, naturally inherited immunity, increase transmission etc. The overview (paper intro) is structured well, and reads at a level where one does not need to be an expert in the field.

The authors then present a mathematical model of disease transmission, that includes both vaccinated and unvaccinated hosts. The authors then go on to use this model to study virulence evolution, given assumptions about LRT and URT protection. When analyzing the model, the authors do not take the standard approach to studying virulence evolution, which would be adaptive dynamics. The authors claim this is because long-term immunity is not well understood in this system. Perhaps waning immunity is a parameter that should be incorporated into the deterministic model (#3), if it is a major assumption in the model analysis.

The authors then calculate R_E (the effective reproductive value). This is essentially R_0 , but not evaluated at the disease-free equilibrium, but at 75% (or less?) of that value (since the authors assume 25% of individuals are already in the C class). At least this is my interpretation of what they are doing. The analysis section is very limited in its explanation. What I believe the authors are doing, is using R_E as a proxy for fitness. R_E acts as the growth rate of a strain (unique alpha) in a population that is smaller than the disease free population. The adaptive dynamics approach works by “competing” strains against one another by calculating invasion fitness. By using R_E , the authors ignore competing existing strains in the population. They then compare the growth rate of a standard B117 strain to that of a different strain (with a different alpha). The issue is, the population is not totally naïve, since 25% have already been infected and are sitting in the C class. This suggests there should be a significant number of infectious individuals in the population, with the B117 infection. Therefore the growth rate of the B117 strain and the novel strain should be calculated with this in mind. I am not convinced that the authors approach represents the dynamics that they claim. Perhaps simulating short term growth rates given the assumptions above is the way to go?

It would be helpful if the authors could provide their analytical expression for R_0 , R_E and any other math used in this analysis (including code), either in the SI or in the main text. The analysis section is very short, and primarily a justification for not using adaptive dynamics.

The authors could include waning immunity into their model and leave it as a free parameter. Then they could take an adaptive dynamics approach and see how varying levels of immunity impact virulence evolution.

Minor Suggestions:

Figure 1. Nice figure. Only issue is in the caption there is discussion of parameter values, well before the model is introduced. Either make these parameters clear or leave them out.

Some changes in font size (see Line 178 ...compartments is known to exist...).

Line 110. Missing “more’ transmissible? (see sentence ...has been identified as being 43-82% transmissible ...)

Line 262 “selection for increased virulence decreases...” Does this mean there is still an increase in virulence, but it is less than in other scenarios?’ Are you referring to the sign of the second derivative of selection? Sentence is awkward.

Line 296 double period “..”

Line 443 “we extend previous work” citation of the previous work please.

Line 456 (Equation 1). define b_1 and b_2 and any constraints on these parameters. For example, I assume $b_2 < 1$, if you want this function saturating. Why do both LRT and URT infections contribute equally to transmission (ie have the same b_1). Does the b_1 parameter need to exist then? Is it important for scaling? What if ϵ is large, then a LRT infection would lead to proportionally more transmission, is this biologically accurate? If $\alpha < 0.0025$, then transmission is negative. What does negative transmission mean biologically, how does this impact the model? What is, or is there, a the relationship between r_U and r_L (Equation 1) and $(r_{\{U,C\}} / r_{\{U,V\}})$ and $(r_{\{L,C\}} / r_{\{L,V\}})$.

Line 508: what do you mean by “more fit” what does this mean mathematically?

Line 510: Just include the bounds in the α_{optimal} range. Listing three values makes it seem like α_{opt} is one of the three options, not bounded between them.

Review form: Reviewer 3

Is the manuscript scientifically sound in its present form?

Yes

Are the interpretations and conclusions justified by the results?

Yes

Is the language acceptable?

Yes

Do you have any ethical concerns with this paper?

No

Have you any concerns about statistical analyses in this paper?

No

Recommendation?

Accept with minor revision (please list in comments)

Comments to the Author(s)

Overall, very insightful and interesting study, especially considering such limited data on virulence-transmission tradeoffs of SARS-CoV-2 and the long-term effects of immunity from vaccines vs. natural immunity being largely unknown. This is certainly a theoretical model that can be built off of and strengthened into a powerful tool as more data emerges. I have relatively minor clarifying questions and comments that could potentially strengthen the paper (see Appendix A). Since many readers of Royal Society Open Science will not necessarily be experts in virulence evolution terminology, it might be prudent to define some of the terms as they are mentioned (virulence, antigenic evolution, etc.).

Decision letter (RSOS-211021.R0)

Dear Mr Miller

The Editors assigned to your paper RSOS-211021 "Assessing the risk of vaccine-driven virulence evolution in SARS-CoV-2" have now received comments from reviewers and would like you to revise the paper in accordance with the reviewer comments and any comments from the Editors. Please note this decision does not guarantee eventual acceptance.

Please submit your revised manuscript and required files (see below) no later than 21 days from today's (ie 04-Oct-2021) date. Note: the ScholarOne system will 'lock' if submission of the revision is attempted 21 or more days after the deadline. If you do not think you will be able to meet this deadline please contact the editorial office immediately.

on behalf of Dr Karen Polizzi (Associate Editor) and Kevin Padian (Subject Editor)
openscience@royalsociety.org

Associate Editor Comments to Author (Dr Karen Polizzi):

We apologise for the length of time that this manuscript was in review. We had difficulty securing the required number of reviewers, but have now finally received three review reports.

As you see, the reviewers were broadly very supportive, but do recommend some changes and additional work on the manuscript. One of these changes is to incorporate the Delta variant into

the paper, which I know you contacted the Office with a desire to do anyway. If there are requested changes that you do not feel are necessary, then please address this in your point-by-point response.

I look forward to receiving your revised manuscript. Please note that it might be useful to add additional files to your Github repository to enable others to use the code. The existing files seem to be missing the source code needed to create the supplementary figure files (or this has been renamed). It is, of course, not a requirement of the journal, but was noted by one of the reviewers. There was also a suggestion to include a bit more information in the README file as a guide.

Reviewer comments to Author:

Reviewer: 1

Comments to the Author(s)

I enjoyed reading Miller and Metcalf's manuscript on assessing the risk of virulence evolution in SARS-CoV-2 as a response to vaccination. It is well written overall, and shows a clear understanding of virulence evolution theory. Given the timeliness of the topic, there is a need for more theory papers addressing evolutionary trajectories of SARS-CoV-2, particularly in light of growing evidence that our most common vaccines are, in fact, more "leaky" than hoped. I enjoyed the attention to detail of the two regions of infection that are linked to transmission (URT) and severe disease (LRT) and appreciate the inclusion of this without having to overly complicate the model.

While the methods are sound overall and the model is simple enough to answer the question, the manuscript is considerably outdated in various ways. I understand covid research moves very quickly but any paper to be posted as a preprint this summer and published this fall should include more recent examples, data, and discussion of the most important VOCs currently under investigation.

There are a couple glaring omissions:

1) There is a lack of references to recent papers on SARS-CoV-2 evolution theory or commentaries by important evolutionary biologists in this subfield. Namely: Day et al. 2021 Current Biology, Otto et al. 2021 Current Biology, and in particular Alizon and Sofonea 2021 J. Evo.Bio. As indicated below, there are several places where these (and some of their citations within) should be cited and the authors results be discussed in the context of these works (e.g. in the discussion section).

2) The Delta variant needs to be included. I understand that this manuscript may have been written at a time before Delta had emerged but unfortunately, this variant is too important evolutionarily to omit. It has both increases in transmission and virulence (e.g. Fisman and Tueite 2021 medrxiv) and reductions in vaccine efficacy. The authors consider cases where the optimum virulence is either intermediate between the ancestral strain and B.1.1.7 or greater than B.1.1.7, however, given how delta is sweeping in various geographical regions, it's pretty clear now that the case where the optimum is below B.1.1.7's virulence is not likely. The question in everyone's mind is, how high can virulence get? Please run the model calibrated for the delta variant's parameters (as those done for B.1.1.7 in lines 495-516) and change figures 3 and 4 to include only B.1.1.7, delta and a hypothetical "more virulent than delta" variant, given that the $\alpha_{\text{optim}} = 1.25 \cdot \alpha_{\text{anc}}$ case is no longer relevant.

Other comments

Fig. 1: The first figure should be a schematic of the transmission-virulence trade-off, particularly showing the different shapes it might have. The section "theory of virulence evolution" is nicely written but it's confusing to use a results image as an illustrative image of the concept (e.g. the unknown parameters and how they are related is confusing given the equations/model have not

been presented yet). A schematic of the trade-off could be added as the first panel to Fig.1 or have a new stand-alone figure in the introduction.

Paragraph ending in line 120: Delta needs to be added here.

Paragraph starting in line 121: please cite and discuss previous published works mentioned number 1 above.

Line 132: this decoupling of death from the trade-off has been discussed in the worked mentioned in number 1, please cite.

Paragraph starting on line 285: please include real-world data here not just the vaccine trials, since we now have estimates for these in global populations.

A direct analysis of vaccine leakiness (how much infectious viral shedding per vaccinated host) and its potential to drive virulence evolution would require a nested model approach, which is, understandably, beyond the scope of this study. However, given parallels of our current vaccine situation with that of Marek's Disease virus (which the authors briefly mention), the most studied example of vaccine-driven virulence evolution, it would be of use to the reader to understand the authors' results in context of this example (e.g. Read et al. 2015 Plos Biology). Please include this in the discussion section.

Points relating to herd immunity:

Vaccine-induced herd immunity is an important part of the results of this paper and given that reaching HIT (herd-immunity threshold) is a moving target for SARS-CoV-2 (e.g. Hodgson et al. 2021 Eurosurveillance), here are some specific points to address about herd immunity.

The underlying R_0 of a virus really affects the needed level of coverage to reach HIT and delta has changed this. How do the authors' results of their scenarios in Fig. 4, such as "how likely herd immunity without virulence evolution", change when considering the delta variant?

Line 306: "... 3) existing vaccines ..." this is still theoretically true but please cite/add any caveats that have been debated in the modelling literature. For example, the vaccines don't reduce transmission as much as we hoped (especially with < 2 doses and for delta) and the projections for vaccine coverages needed for reaching herd immunity are changing (e.g. Mancuso, Eikenberry and Gumel 2021 medrxiv).

In figures 3 and 4, the 10% vaccinated coverage panels are there for completeness but are not really what we are aiming for globally. It would make more sense to just present two columns of 50 and 90 or have a gradation from 50 to 100%.

Reviewer: 2

Comments to the Author(s)

The authors provide a well-written overview of how SARS-CoV-2 might evolve towards greater virulence in response to numerous factors, such as vaccination, naturally inherited immunity, increase transmission etc. The overview (paper intro) is structured well, and reads at a level where one does not need to be an expert in the field.

The authors then present a mathematical model of disease transmission, that includes both vaccinated and unvaccinated hosts. The authors then go on to use this model to study virulence evolution, given assumptions about LRT and URT protection. When analyzing the model, the authors do not take the standard approach to studying virulence evolution, which would be

adaptive dynamics. The authors claim this is because long-term immunity is not well understood in this system. Perhaps waning immunity is a parameter that should be incorporated into the deterministic model (#3), if it is a major assumption in the model analysis.

The authors then calculate R_E (the effective reproductive value). This is essentially R_0 , but not evaluated at the disease-free equilibrium, but at 75% (or less?) of that value (since the authors assume 25% of individuals are already in the C class). At least this is my interpretation of what they are doing. The analysis section is very limited in its explanation. What I believe the authors are doing, is using R_E as a proxy for fitness. R_E acts as the growth rate of a strain (unique α) in a population that is smaller than the disease free population. The adaptive dynamics approach works by “competing” strains against one another by calculating invasion fitness. By using R_E , the authors ignore competing existing strains in the population. They then compare the growth rate of a standard B117 strain to that of a different strain (with a different α). The issue is, the population is not totally naïve, since 25% have already been infected and are sitting in the C class. This suggests there should be a significant number of infectious individuals in the population, with the B117 infection. Therefore the growth rate of the B117 strain and the novel strain should be calculated with this in mind. I am not convinced that the authors approach represents the dynamics that they claim. Perhaps simulating short term growth rates given the assumptions above is the way to go?

It would be helpful if the authors could provide their analytical expression for R_0 , R_E and any other math used in this analysis (including code), either in the SI or in the main text. The analysis section is very short, and primarily a justification for not using adaptive dynamics.

The authors could include waning immunity into their model and leave it as a free parameter. Then they could take an adaptive dynamics approach and see how varying levels of immunity impact virulence evolution.

Minor Suggestions:

Figure 1. Nice figure. Only issue is in the caption there is discussion of parameter values, well before the model is introduced. Either make these parameters clear or leave them out.

Some changes in font size (see Line 178 ...compartments is known to exist...).

Line 110. Missing “more’ transmissible? (see sentence ...has been identified as being 43-82% transmissible ...)

Line 262 “selection for increased virulence decreases...” Does this mean there is still an increase in virulence, but it is less than in other scenarios? Are you referring to the sign of the second derivative of selection? Sentence is awkward.

Line 296 double period “..”

Line 443 “we extend previous work” citation of the previous work please.

Line 456 (Equation 1). define b_1 and b_2 and any constraints on these parameters. For example, I assume $b_2 < 1$, if you want this function saturating. Why do both LRT and URT infections contribute equally to transmission (ie have the same b_1). Does the b_1 parameter need to exist then? Is it important for scaling? What if ϵ is large, then a LRT infection would lead to proportionally more transmission, is this biologically accurate? If $\alpha < 0.0025$, then transmission is negative. What does negative transmission mean biologically, how does this

impact the model? What is, or is there, a the relationship between r_U and r_L (Equation 1) and $(r_{\{U,C\}} / r_{\{U,V\}})$ and $(r_{\{L,C\}} / r_{\{L,V\}})$.

Line 508: what do you mean by “more fit” what does this mean mathematically?

Line 510: Just include the bounds in the α_{optimal} range. Listing three values makes it seem like α_{opt} is one of the three options, not bounded between them.

Reviewer: 3

Comments to the Author(s)

Overall, very insightful and interesting study, especially considering such limited data on virulence-transmission tradeoffs of SARS-CoV-2 and the long-term effects of immunity from vaccines vs. natural immunity being largely unknown. This is certainly a theoretical model that can be built off of and strengthened into a powerful tool as more data emerges. I have relatively minor clarifying questions and comments that could potentially strengthen the paper. Since many readers of Royal Society Open Science will not necessarily be experts in virulence evolution terminology, it might be prudent to define some of the terms as they are mentioned (virulence, antigenic evolution, etc.) (see attached file: "Miller_Metcalf.pdf").

===PREPARING YOUR MANUSCRIPT===

===PREPARING YOUR REVISION IN SCHOLARONE===

Author's Response to Decision Letter for (RSOS-211021.R0)

See Appendix B.

Decision letter (RSOS-211021.R1)

Dear Mr Miller,

It is a pleasure to accept your manuscript entitled "Assessing the risk of vaccine-driven virulence evolution in SARS-CoV-2" in its current form for publication in Royal Society Open Science. The comments of the reviewer(s) who reviewed your manuscript are included at the foot of this letter.

Please remember to make any data sets or code libraries 'live' prior to publication, and update any links as needed when you receive a proof to check - for instance, from a private 'for review' URL to a publicly accessible 'for publication' URL. It is good practice to also add data sets, code and other digital materials to your reference list. In particular, we ask that you please archive your GitHub code within the Zenodo repository: <https://guides.github.com/activities/citable-code/>. By doing this, a formal, citable DOI will be associated with your data record, and an open license (CC-BY preferred) can be applied to your data. We would then ask that you please update your data availability statement to read as:

"Data and relevant code for this research work are stored in GitHub: [GitHub URL here] and have been archived within the Zenodo repository: <https://doi.org/zenodo.....> [ref number].

COVID-19 rapid publication process:

We are taking steps to expedite the publication of research relevant to the pandemic. If you wish, you can opt to have your paper published as soon as it is ready, rather than waiting for it to be published the scheduled Wednesday.

This means your paper will not be included in the weekly media round-up which the Society sends to journalists ahead of publication. However, it will still appear in the COVID-19 Publishing Collection which journalists will be directed to each week (<https://royalsocietypublishing.org/topic/special-collections/novel-coronavirus-outbreak>).

If you wish to have your paper considered for immediate publication, or to discuss further, please notify openscience_proofs@royalsociety.org and press@royalsociety.org when you respond to this email.

on behalf of Dr Karen Polizzi (Associate Editor) and Kevin Padian (Subject Editor)
openscience@royalsociety.org

Associate Editor Comments to Author (Dr Karen Polizzi):
Associate Editor

Comments to the Author:

The revised version satisfactorily addresses the reviewers' comments including a broader discussion of the picture given the emergence of delta since the original manuscript was submitted. The broadened discussion on waning of immunity is very useful. Overall, this work is a novel contribution to our understanding of the potential for SARS-CoV2 evolution. It is written in an accessible way and the conclusions are supported by the modelling.

Reviewer comments to Author:

Follow Royal Society Publishing on Twitter: [@RSocPublishing](https://twitter.com/RSocPublishing)

Appendix A

Overall, very insightful and interesting study, especially considering such limited data on virulence-transmission tradeoffs of SARS-CoV-2 and the long-term effects of immunity from vaccines vs. natural immunity being largely unknown. This is certainly a theoretical model that can be built off of and strengthened into a powerful tool as more data emerges. I have relatively minor clarifying questions and comments that could potentially strengthen the paper, listed below. Since many readers of Royal Society Open Science will not necessarily be experts in virulence evolution terminology, it might be prudent to define some of the terms as they are mentioned (virulence, antigenic evolution, etc.).

22: In the first sentence, would be good to put "as of September 2021" as a reference, since the number of cases and deaths may increase in time.

45-46: Reference for "quantitative virulence" definition?

45: It might be useful here to define virulence, as various scientists have different definitions (i.e., are you defining it as host-induced mortality, or something else?)

46-48: Do you have a reference for the sentence, "In the absence of any associated costs..."?

70-73: Cite an example of a disease truncating the infectious period (perhaps in a case other than death) and a trade-off emerging?

110: Here, do you mean to say 43-82% **more** transmissible?

111: How do founder effects shape estimates of transmissibility?

157: I think you meant "directionality" instead of "directionally."

152, 157: Speaking of which, what do you mean by "directionality of selection for increased virulence"? Since we know the directionality (increasing virulence), wouldn't it be more accurate to say the "...strength of selection for increased virulence..."? If you mean to say the directionality of selection for increased virulence being either selecting **for** increased virulence or selecting **against** an increase in virulence, that might be good to clarify here. I wasn't sure what you meant by "directionality" until I saw it more clearly in Figure 3's "change in selection" color bar on the right side of the figure.

168-170: The sentence beginning with "Vaccinal immunity..." is quite confusing to me. Can you rephrase? I think you go on to explain what you mean in the subsequent sentences. ...Vaccines can reduce illness without reducing transmission to the same extent. Because transmission is not reduced, virulence could increase. But virulence could only increase if transmission increases and if virulence does not have an impact on mortality. Is that a correct interpretation of that sentence?

203-207: About the study mentioned here: Were all the healthcare workers vaccinated? Or had they been previously infected and then confirmed to have IgG antibodies? For the seronegative individuals, they became symptomatic after they were tested, even though they were shown to not have any detectable virus?

Figure 2A: This diagram is a bit confusing to me. Do the black vs. green arrows mean anything? At first, I thought that a green arrow represented a decrease in transmission/infection/severity, and the black meant an increase, but they all

represent increases? (Nor is the color of the arrow representative of the location of infection/vaccination (URT vs. LRT?)

301: Here, I think you mean to say, "the emergence of vaccine resistance."

362-363: Do we have data on how each of the vaccines listed in lines 285-297 differ in their abilities to differentially protect against disease and transmission?

Figure 4: Just a thought: I'm curious to know what these dynamics would look like if accounting for those currently fully vaccinated in the United States (56.2% as of Sept. 2020), versus the world (33%), and based on the most common circulating genotype. (Obviously somewhere in between the 50%-90% plots for the former and between 10-50% for the latter but would still be neat to see the simulation plots for those specific, real-world numbers.)

355: Do you think there is a "deadline" to get to a threshold of vaccinations to ensure herd immunity? I.e., if we still have 60% vaccinated population by this time next year, does that change anything about herd immunity and/or whether a virulent variant may emerge?

395: Switch the words "the" and "both" to "both the effective population..."

398-399: Waning immunity is a very good point and a big unknown that could radically change predicted outcomes. Good point to have here.

405: This data is probably being collected by public health agencies? (I hope)

417: Some readers may think at first you are discussing host resistance, so perhaps to make it clearer, you could state "i.e., viral resistance to the vaccine", or something similar.

542: Citation (Glennon et al. Epidemics) in (number) format.

543: Since it is now nearly October 2020, have the numbers (25% of individuals convalescent) changed over the past 10 months? If so, does that affect the model predictions?

Supplemental figures not included in manuscript draft?

Appendix B

Response to reviewer comments

Assessing the risk of vaccine-driven virulence evolution in SARS-CoV-2

Authors: Ian F. Miller, C. Jessica E. Metcalf

Dear Editor and reviewers,

We appreciate the very thoughtful and detailed comments you have provided us. We have made substantial updates to the paper in light of the emergence of the *delta* variant. Additionally, we have switched our analysis methods to a more standard adaptive dynamics approach that is aimed more at predicting trajectories of SARS-CoV-2 evolution than at identifying immediate patterns of selection for changes in virulence. These changes have had a minimal impact on our conclusions and closing recommendations.

Please find a point by point response to reviewer comments below.

Sincerely,

Ian Miller and Jess Metcalf

Reviewer 1:

1. There is a lack of references to recent papers on SARS-CoV-2 evolution theory or commentaries by important evolutionary biologists in this subfield. Namely: Day et al. 2021 Current Biology, Otto et al. 2021 Current Biology, and in particular Alizon and Sofonea 2021 J. Evo.Bio. As indicated below, there are several places where these (and some of their citations within) should be cited and the authors results be discussed in the context of these works (e.g. in the discussion section).

We added citations to these works throughout the paper where appropriate and added a line in the discussion stating: ‘This work builds upon the theoretical analyses of SARS-CoV-2 virulence evolution presented in many previous studies by extending the same fundamental concepts to the issue of vaccine-driven evolution.’ (Line 606)

2. The Delta variant needs to be included. I understand that this manuscript may have been written at a time before Delta had emerged but unfortunately, this variant is too important evolutionarily to omit. It has both increases in transmission and virulence (e.g. Fisman and Tueite 2021 medrxiv) and reductions in vaccine efficacy. The authors consider cases where the optimum virulence is either intermediate between the ancestral strain and B.1.1.7 or greater than B.1.1.7, however, given how delta is sweeping in various geographical regions, it’s pretty clear now that the case where the optimum is below B.1.1.7’s virulence is not likely. The question in everyone’s mind is, how high can virulence get? Please run the model calibrated for the delta variant’s parameters (as those done for B.1.1.7 in lines 495-516) and change figures 3 and 4 to include only B.1.1.7, delta and a hypothetical “more virulent than delta” variant, given that the $\alpha_{\text{optim}} = 1.25 * \alpha_{\text{ansc}}$ case is no longer relevant.

We strongly agree that current evidence surrounding *delta* necessitates a reconsideration of the optimum virulence scenarios. We changed these to reflect the possibility that optimum virulence could 1) fall between that of *alpha* and *delta*, 2) be equal to that of *delta*, or 3) be greater than that of *delta* (See methods and new figure S1). We also expanded the introduction and discussion to include details about delta.

3. Fig. 1: The first figure should be a schematic of the transmission-virulence trade-off, particularly showing the different shapes it might have. The section ‘theory of virulence evolution’ is nicely written but it’s confusing to use a results image as an illustrative image of the concept (e.g. the unknown parameters and how they are related is confusing given the equations/model have not been presented yet). A schematic of the trade-off could be added as the first panel to Fig.1 or have a new stand-alone figure in the introduction.

We agree that our paper could be improved by a figure more focused on the potential forms of the relationship between virulence and transmission. We created a new figure

(Fig. 1) and moved the original Fig. 1 to the supplement (Fig. S1) while also updating it to reflect the changes detailed in point 2 above.

4. Paragraph ending in line 120: Delta needs to be added here.

We added a discussion of delta to this section (beginning at L140).

5. Paragraph starting in line 121: please cite and discuss previous published works mentioned number 1 above.

We added a discussion of these works to this section (L211-247).

6. Line 132: this decoupling of death from the trade-off has been discussed in the worked mentioned in number 1, please cite.

We added this citation as suggested (L221).

7. Paragraph starting on line 285: please include real-world data here not just the vaccine trials, since we now have estimates for these in global populations.

We have updated these paragraphs with current evidence regarding real-world vaccine efficacy and *delta* (L457-506).

8. A direct analysis of vaccine leakiness (how much infectious viral shedding per vaccinated host) and its potential to drive virulence evolution would require a nested model approach, which is, understandably, beyond the scope of this study. However, given parallels of our current vaccine situation with that of Marek's Disease virus (which the authors briefly mention), the most studied example of vaccine-driven virulence evolution, it would be of use to the reader to understand the authors' results in context of this example (e.g. Read et al. 2015 Plos Biology). Please include this in the discussion section.

We added context about how the results of the MDV study relate to COVID in lines 272-276: "It is important to note that some factors present in these examples of vaccine-driven virulence evolution are absent in the case of SARS-CoV-2. For instance, in the latter example, viral virulence was extremely high (60-100% infection fatality rate) prior to vaccine-driven evolution, and vaccinal protection against infection was extremely poor."

9. Vaccine-induced herd immunity is an import part of the results of this paper and given that reaching HIT (herd-immunity threshold) is a moving target for SARS-CoV-2 (e.g. Hodgson et al. 2021 Eurosurveillance), here are some specific points to address about herd immunity.
 - a. The underlying R0 of a virus really affects the needed level of coverage to reach HIT and delta has changed this. How does the authors' results of their scenarios in

Fig. 4, such as “how likely herd immunity without virulence evolution”, change when considering the delta variant?

We updated the parameterization of b_1 and b_2 to reflect a reasonable value of R_0 for *delta* (L977).

- b. Line 306: “... 3) existing vaccines ...” this is still theoretically true but please cite/add any caveats that have been debated in the modelling literature. For example, the vaccines don’t reduce transmission as much as we hoped (especially with < 2 doses and for *delta*) and the projections for vaccine coverages needed for reaching herd immunity are changing (e.g. Mancuso, Eikenberry and Gumel 2021 medrxiv).

We added the caveat “although waning vaccinal protection could jeopardize this outcome” at line 514. We also added a detailed discussion of what herd immunity implies for COVID transmission (i.e. not necessarily a ‘COVID-free’ world) and how it relates to waning (L609-656).

- c. In figures 3 and 4, the 10% vaccinated coverage panels are there for completeness but are not really what we are aiming for globally. It would make more sense to just present two columns of 50 and 90 or have a gradation from 50 to 100%.

We updated the vaccine coverages considered in our analyses to 50%, 75%, and 90%.

Reviewer 2:

1. The authors provide a well-written overview of how SARS-CoV-2 might evolve towards greater virulence in response to numerous factors, such as vaccination, naturally inherited immunity, increase transmission etc. The overview (paper intro) is structured well, and reads at a level where one does not need to be an expert in the field.

The authors then present a mathematical model of disease transmission, that includes both vaccinated and unvaccinated hosts. The authors then go on to use this model to study virulence evolution, given assumptions about LRT and URT protection. When analyzing the model, the authors do not take the standard approach to studying virulence evolution, which would be adaptive dynamics. The authors claim this is because long-term immunity is not well understood in this system. Perhaps waning immunity is a parameter that should be incorporated into the deterministic model (#3), if it is a major assumption in the model analysis.

We changed our methods to a more formal adaptive dynamics approach. Waning immunity will certainly be important for SARS-CoV-2 epidemiological dynamics, and potentially evolutionary dynamics as well. However, as we have witnessed over the past year, significant amounts of virulence evolution can occur in a matter of months. Incorporating waning that leads to a total loss of immunity would mean that

epidemiological dynamics would take many years to reach equilibrium, forcing predictions about virulence evolution to fall out of step with the observed pace of virulence evolution. However, we have also witnessed that immunity has waned. Evidence suggests that it weakened rather than disappeared, as protection against severe disease remains robust. Our model formulation can account for this if the r parameters are interpreted as population means, summarizing the variation in immunity across all individuals in each class. We discuss these points in lines 994 to 1010 and elsewhere. To make our model as useful as possible, we have also included waning parameters (set to 0 in our analyses) so that waning can be investigated in the future if appropriate.

2. The authors then calculate R_E (the effective reproductive value). This is essentially R_0 , but not evaluated at the disease-free equilibrium, but at 75% (or less?) of that value (since the authors assume 25% of individuals are already in the C class). At least this is my interpretation of what they are doing. The analysis section is very limited in its explanation. What I believe the authors are doing, is using R_E as a proxy for fitness. R_E acts as the growth rate of a strain (unique α) in a population that is smaller than the disease free population. The adaptive dynamics approach works by “competing” strains against one another by calculating invasion fitness. By using R_E , the authors ignore competing existing strains in the population. They then compare the growth rate of a standard B117 strain to that of a different strain (with a different α). The issue is, the population is not totally naïve, since 25% have already been infected and are sitting in the C class. This suggests there should be a significant number of infectious individuals in the population, with the B117 infection. Therefore the growth rate of the B117 strain and the novel strain should be calculated with this in mind. I am not convinced that the authors approach represents the dynamics that they claim. Perhaps simulating short term growth rates given the assumptions above is the way to go?

We have added more details to the explanation of our calculation of basic and effective reproductive numbers and how they relate to the frequencies of individuals in each model class in equation 4 and lines 975 and 1069. Our approach now conforms to a typical adaptive dynamics framework, which resolves the concerns raised here.

3. It would be helpful if the authors could provide their analytical expression for R_0 , R_E and any other math used in this analysis (including code), either in the SI or in the main text. The analysis section is very short, and primarily a justification for not using adaptive dynamics.

We added more details about the calculation of these values (see point 2 above). Due to the complexity of the model, a simple closed form expression for R is not available. Code for all analyses can be found in reference 81.

4. The authors could include waning immunity into their model and leave it as a free parameter. Then they could take an adaptive dynamics approach and see how varying levels of immunity impact virulence evolution.

See point 1 above.

5. Figure 1. Nice figure. Only issue is in the caption there is discussion of parameter values, well before the model is introduced. Either make these parameters clear or leave them out.

Thank you. We revised Fig. 1 to illustrate various shapes of the relationship between virulence and transmission. We moved the original Fig. 1 to the supplement (Fig. S1), and made that figure's purpose purely to illustrate assumptions about the optimum value of virulence.

6. Some changes in font size (see Line 178 ...compartments is known to exist...).

We were unable to identify any font size discrepancy in the word document version of our manuscript. Perhaps this problem arose during PDF generation. We will keep a look out for this problem moving forward.

7. Line 110. Missing "more" transmissible? (see sentence ...has been identified as being 43-82% transmissible ...)

We added the missing "more" at line 150.

8. Line 262 "selection for increased virulence decreases..." Does this mean there is still an increase in virulence, but it is less than in other scenarios? Are you referring to the sign of the second derivative of selection? Sentence is awkward.

This wording is no longer present in our description of the results of our adaptive dynamics analysis.

9. Line 296 double period "..."

We removed the double period.

10. Line 443 "we extend previous work" citation of the previous work please.

We changed this phrasing to "We separate the impacts...' (L798), as this feature of our analysis is novel.

11. Line 456 (Equation 1). define b_1 and b_2 and any constraints on these parameters. For example, I assume $b_2 < 1$, if you want this function saturating. Why do both LRT and URT infections contribute equally to transmission (ie have the same b_1). Does the b_1 parameter need to exist then? Is it important for scaling? What if epsilon is large, then a LRT infection would lead to proportionally more transmission, is this biologically accurate? If $\alpha < 0.0025$, then transmission is negative. What does negative transmission mean biologically, how does this impact the model? What is, or is there, a the relationship between r_U and r_L (Equation 1) and $(r_{\{U,C\}} / r_{\{U,V\}})$ and $(r_{\{L,C\}} / r_{\{L,V\}})$.

We clarify that b_1 and b_2 are simply shape parameters in line 803. The b_1 parameter is important for setting the correct value of R_0 , and has no effect on setting the balance of transmission between the URT and LRT. Values of $b_2 < 1$ do indeed correspond to saturating relationships. We do not consider infections to be either LRT or URT. Rather, ϵ sets the fraction of total onwards transmission generated by each compartment. We do not consider α values less than 0.0025 in our analysis. Values less than 0.0025 would indeed correspond to negative transmission, but this is a trivial, biologically meaningless case. As outlined in the methods section, we vary the values of r_{Uv} , and r_{Lv} , and we fix the values of r_{Uc} and r_{Lc} . We set the value of $r_{Uc,v}$ and $r_{Lc,v}$ depending on the values of r_{Uv} and r_{Lv} .

12. Line 508: what do you mean by “more fit” what does this mean mathematically?

We added text in line 948 clarifying that this assumption corresponds to the assumed values of R_0 .

13. Line 510: Just include the bounds in the α_{optimal} range. Listing three values makes it seem like α_{opt} is one of the three options, not bounded between them.

The optimum α value is indeed just one of the three values. See Fig. S1, Fig 3, methods section for how this plays out in the results of our analyses.

Reviewer 3:

1. Overall, very insightful and interesting study, especially considering such limited data on virulence-transmission tradeoffs of SARS-CoV-2 and the long-term effects of immunity from vaccines vs. natural immunity being largely unknown. This is certainly a theoretical model that can be built off of and strengthened into a powerful tool as more data emerges. I have relatively minor clarifying questions and comments that could potentially strengthen the paper, listed below. Since many readers of Royal Society Open Science will not necessarily be experts in virulence evolution terminology, it might be prudent to define some of the terms as they are mentioned (virulence, antigenic evolution, etc.).
2. Line 22: In the first sentence, would be good to put “as of September 2021” as a reference, since the number of cases and deaths may increase in time.

We updated these numbers, and added in “as of October 2021” (Line 37-38).

3. Lines 45-46: Reference for “quantitative virulence” definition?

We changed this term to simply “virulence”. We sometimes differentiate between quantitative and qualitative virulence because some in the world of plant disease ‘virulence’ is used to describe the ability to infect rather than the fitness costs of infection. But for a paper clearly addressing virulence evolution in humans, this differentiation is not necessary.

4. Line 45: It might be useful here to define virulence, as various scientists have different definitions (i.e., are you defining it as host-induced mortality, or something else?)

We added more details to our definition of virulence (L71-72).

5. Line 46-48: Do you have a reference for the sentence, “In the absence of any associated costs...”?

This statement is supported by references 10-12 later in the paragraph.

6. Line 70-73: Cite an example of a disease truncating the infectious period (perhaps in a case other than death) and a trade-off emerging?

We prefer to not cite an example here, as the argument we are making is based in logic rather than specific pieces of evidence.

7. Line 110: Here, do you mean to say 43-82% more transmissible?

Yes, see point 7 in response to viewer 2 above.

8. Line 111: How do founder effects shape estimates of transmissibility?

We removed the caveat about founder effects from the text. At the time when we originally wrote this section, there was more uncertainty about the transmissibility of alpha, and we felt that a caveat was appropriate. We no longer think that this is necessary. Founder effects can stochastically lead to the proliferation of strains and as a result make them appear more transmissible. This is discussed in Davies et al. (2021), although the authors argue that it is likely not the case for alpha.

9. Line 157: I think you meant “directionality” instead of “directionally.”

We corrected this word.

10. Lines 152, 157: Speaking of which, what do you mean by “directionality of selection for increased virulence”? Since we know the directionality (increasing virulence), wouldn't it be more accurate to say the “...strength of selection for increased virulence...”? If you mean to say the directionality of selection for increased virulence being either selecting for increased virulence or selecting against an increase in virulence, that might be good to clarify here. I wasn't sure what you meant by “directionality” until I saw it more clearly in Figure 3's “change in selection” color bar on the right side of the figure.

This point no longer applies to our new set of results which focus on evolutionarily stable virulence strategies rather than on selection.

11. Lines 168-170: The sentence beginning with “Vaccinal immunity...” is quite confusing to me. Can you rephrase? I think you go on to explain what you mean in the subsequent sentences. ...Vaccines can reduce illness without reducing transmission to the same extent. Because transmission is not reduced, virulence could increase. But virulence could only increase if transmission increases and if virulence does not have an impact on mortality. Is that a correct interpretation of that sentence?

We agree that our original phrasing was confusing. We changed it to “Vaccinal immunity to SARS-CoV-2 could open the door for virulence evolution, if reduces disease to a greater extent than transmission.” (Line 278).

12. Lines 203-207: About the study mentioned here: Were all the healthcare workers vaccinated? Or had they been previously infected and then confirmed to have IgG antibodies? For the seronegative individuals, they became symptomatic after they were tested, even though they were shown to not have any detectable virus?

We clarified that the healthcare workers were unvaccinated and that those with IgG antibodies acquired them through natural infection (L326). Seronegative individuals are those who do not have detectable antibodies that would be consistent with previous infection.

13. Figure 2A: This diagram is a bit confusing to me. Do the black vs. green arrows mean anything? At first, I thought that a green arrow represented a decrease in transmission/infection/severity, and the black meant an increase, but they all represent increases? (Nor is the color of the arrow representative of the location of infection/vaccination (URT vs. LRT)?)

The black and green arrows are labeled, and colors+labels are consistent between panels A and B.

14. Line 301: Here, I think you mean to say, “the emergence of vaccine resistance.”

You are correct. We corrected this text in line 510.

15. Lines 362-363: Do we have data on how each of the vaccines listed in lines 285-297 differ in their abilities to differentially protect against disease and transmission?

We added new text that describes current evidence surrounding vaccine efficacy (L457-506).

16. Figure 4: Just a thought: I’m curious to know what these dynamics would look like if accounting for those currently fully vaccinated in the United States (56.2% as of Sept. 2020), versus the world (33%), and based on the most common circulating genotype. (Obviously somewhere in between the 50%-90% plots for the former and between 10-50% for the latter but would still be neat to see the simulation plots for those specific, real-world numbers.)

We prefer to keep our analyses focused on bounding the problem of virulence evolution. Given the amount of uncertainty surrounding potential for virulence evolution, we believe that trying to make pinpoint projections would be more misleading than productive.

17. Line 355: Do you think there is a “deadline” to get to a threshold of vaccinations to ensure herd immunity? I.e., if we still have 60% vaccinated population by this time next year, does that change anything about herd immunity and/or whether a virulent variant may emerge?

Predicting the pace of virulence evolution is even more difficult than predicting the trajectory of evolution, and unfortunately we cannot say anything about a deadline at this time.

18. Line 395: Switch the words “the” and “both” to “both the effective population...”

We fixed this error.

19. Lines 398-399: Waning immunity is a very good point and a big unknown that could radically change predicted outcomes. Good point to have here.

Thank you. We added additional discussion of waning immunity throughout the paper in response to points raised by other reviewers.

20. Lines 405: This data is probably being collected by public health agencies? (I hope)

Yes, thankfully variants that may be more virulent are monitored carefully. We changed this wording to “continue to be surveilled” (L728). We also removed the need for monitoring from the abstract since this is now happening sufficiently. We now highlight the need for 1) vaccines that provide robust protection against both disease and transmission, and 2) high vaccine coverage rather than monitoring efforts.

21. Line 417: Some readers may think at first you are discussing host resistance, so perhaps to make it clearer, you could state “i.e., viral resistance to the vaccine”, or something similar.

We agree that this text was unclear. We removed our discussion of the evolution of vaccine escape because we felt it was no longer adequate given that VOCs that evade immunity have indeed been identified. This is still an extremely important issue, but one that we feel is now outside the scope of this paper.

22. Line 542: Citation (Glennon et al. Epidemics) in (number) format.

Thank you for pointing out this error. This citation was removed during our switch to an adaptive dynamics approach.

23. Line 543: Since it is now nearly October 2020, have the numbers (25% of individuals convalescent) changed over the past 10 months? If so, does that affect the model predictions?

We updated the initial conditions of our model to reflect the increase in the frequency of convalescent individuals (equation 5).

24. Supplemental figures not included in manuscript draft?

We apologize that these were not included. We submitted them along with the rest of the manuscript, so we believe that this is likely an issue with the manuscript submission system.